# A Classification View on Meta Learning Bandits

**Mirco Mutti** [1]   **Jeongyeol Kwon** [2]   **Shie Mannor** [1 3]   **Aviv Tamar** [1]

## Abstract

Contextual multi-armed bandits are a popular choice to model sequential decision-making. *E.g.*, in a healthcare application we may perform various tests to asses a patient condition (exploration) and then decide on the best treatment to give (exploitation). When humans design strategies, they aim for the exploration to be *fast*, since the patient's health is at stake, and easy to *interpret* for a physician overseeing the process. However, common bandit algorithms are nothing like that: The regret caused by exploration scales with $\sqrt{H}$ over $H$ rounds and decision strategies are based on opaque statistical considerations. In this paper, we use an original *classification view* to meta learn interpretable and fast exploration plans for a fixed collection of bandits $\mathbb{M}$. The plan is prescribed by an interpretable *decision tree* probing decisions' payoff to classify the test bandit. The test regret of the plan in the *stochastic* and *contextual* setting scales with $\mathcal{O}(\lambda^{-2} C_\lambda(\mathbb{M}) \log^2(MH))$, being $M$ the size of $\mathbb{M}$, $\lambda$ a separation parameter over the bandits, and $C_\lambda(\mathbb{M})$ a novel *classification-coefficient* that fundamentally links meta learning bandits with classification. Through a nearly matching lower bound, we show that $C_\lambda(\mathbb{M})$ inherently captures the complexity of the setting.

## 1. Introduction

In the *Multi-Armed Bandits* model (MAB, Lattimore & Szepesvári, 2020), a decision-maker, called the *agent*, faces a collection of unknown probability distributions over reals, called *arms*, representing alternative decisions and their corresponding payoff (a.k.a. *reward*), which the agent repeatedly takes, or *pulls*, to maximize the mean cumulative reward collected over time. In some settings, called *contextual* MABs (Audibert & Bubeck, 2010), the reward of an

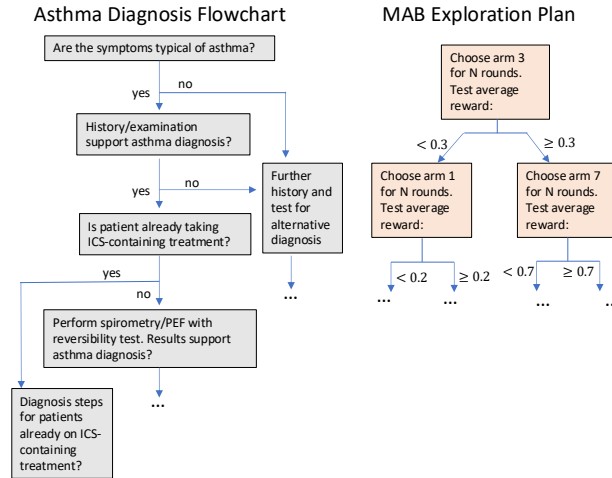

Figure 1: Left: An excerpt from a clinical flowchart for the medical diagnosis of Asthma (GIf, 2023). Right: Illustration of an interpretable exploration plan for a MAB.

arm depends also on a *context*, a vector of features that the agent observes before deciding which arm to pull. The main challenge in MABs is how to pull arms in a way that effectively balances information gathering (called *exploration*) and immediate rewards (called *exploitation*).

A multitude of decision-making problems, ranging from recommender systems (Li et al., 2010) to treatment allocation (Berry, 1978), pricing of goods (Rothschild, 1974), advertising (Schwartz et al., 2017), can be modelled as MAB problems. However, although the problem structure is fitting, typical MAB algorithms are often very different from human-designed decision plans. For example, consider the clinical diagnosis plan illustrated in Figure 1 (left). In machine learning parlance, this plan takes several exploration actions (diagnosis tests) to yield a diagnosis, which will later be treated by appropriate medical actions (exploitation). It is clear that (i) the plan is *short* – fast diagnosis is imperative; and (ii) the plan is *interpretable*, and can be easily communicated both to physicians and patients. Our goal in this work is to develop a framework for short and interpretable action plans in the setting of MABs.

To this end, we consider the *stochastic* contextual MAB formulation, a model of non-adversarial problems whose

*Equal contribution [1]Technion – Israel Institute of Technology [2]University of Wisconsin-Madison [3]NVIDIA Research. Correspondence to: Mirco Mutti <mirco.m@technion.ac.il>.

*Proceedings of the 42nd International Conference on Machine Learning*, Vancouver, Canada. PMLR 267, 2025. Copyright 2025 by the author(s).

theoretical barriers are well-understood (Lai & Robbins, 1985; Auer et al., 2002). Even when the context is fixed, the *regret* the agent has to pay, defined as the difference between the cumulative reward of their decisions and those of the optimal strategy, inevitably scales with $\sqrt{KH}$ in the worst case, being $H$ and $K$ the number of pulls and arms respectively. The latter rate might not be compelling enough in settings in which the regret translates to money losses, such as in pricing or advertising scenarios, or even a negative impact on a patient's health condition, like in the clinical diagnosis problem mentioned above.

Faster performance is possible when prior knowledge about the *class* of bandits the agent faces may be available, such as from historical data or powerful simulators. For example, Thompson sampling (Thompson, 1933) allows to exploit a prior distribution over the problem parameters through a Bayesian-inspired approach. In favorable circumstances, the latter yields an *average* regret rate that is at most logarithmic in the number of arms $K$ (Russo & Van Roy, 2016). Another formulation, called *latent bandits* (Maillard & Mannor, 2014; Hong et al., 2020a), assumes that the problem parameters are coming from a finite collection of bandits. The latter allows to trade a factor of $\sqrt{K}$ with $\sqrt{M}$ in the regret, being $M$ the number of bandits in the collection.

Here we consider a *meta learning* version of latent bandits. We can interact with the collection of bandits to meta-train an algorithm that is then tested against one bandit in the collection, whose identity is not revealed to the algorithm. Unfortunately, any prior knowledge we can extract at meta training cannot improve the $\sqrt{MH}$ rate in the *worst case*, which holds even for a collection of two bandits (Lattimore & Szepesvári, 2020). This changes when we assume that the bandits in the collection are meaningfully different, *i.e.*, the reward distribution of their arms have some statistical *separation* (Chen et al., 2022b; Mutti & Tamar, 2024). The separation condition is relevant in practice: If two patients do not respond differently to at least one treatment, there is little point in modeling them with different bandits. Whereas this can help achieving fast rates, previous work, either with or without separation, do not yield interpretable plans.

To design interpretable exploration plans for bandits, our main technical contribution is connecting ideas from the classification literature to MAB analysis. In principle, the idea is to take advantage of separation to explicitly *classify* the test task from data with high probability, and then exploit the optimal strategy for the classified task. This *classification view* allows to break the common barriers for meta learning bandits, while providing an elegant and original characterization of the regret dynamics under separation.

The contributions of the paper are organized as follows. In Section 2, we describe problem of meta learning bandits and the separation condition. In Section 3, we formalize the classification view of MABs by introducing a novel measure of complexity, the *classification-coefficient* $C_\lambda(\mathbb{M})$ for a $\lambda$-separated set of bandits $\mathbb{M}$ and a space of tests $\Pi_{\mathcal{C}}$, which captures the hardness of the learning problem: When $\mathbb{M}$ is known, a simple *Explicit Classify then Exploit* (ECE) procedure, which runs a classification algorithm to classify the test task and then exploits the optimal policy of the classified task, achieves a test regret of $\mathcal{O}(\lambda^{-2}C_\lambda(\mathbb{M})\log^2(MH))$ over $H$ rounds. Through a sample complexity lower bound to identify the optimal policy at test time, we show that the factor $\lambda^{-2}C_\lambda(\mathbb{M})$ is indeed unavoidable in the worst case. In Section 4, we provide a practical implementation of ECE with *decision trees* – a standard tool in interpretable decision making (Bressan et al., 2024) – that nearly matches the regret above while yielding a fully interpretable exploration plan (like in Figure 1 right). The latter is robust to mis-specifications of $\mathbb{M}$, which is estimated through a tractable meta training routine. Notably, all of our results hold for the contextual setting. Section 5 provides numerical experiments that showcase our algorithms against UCB/TS-like approaches for latent bandits (Hong et al., 2020a). Section 6 is dedicated to related works. The proofs of the theorems are in the appendix.

## 2. Problem setting

Let us consider a finite collection of contextual bandit problems $\mathbb{M} := \{\nu_i\}_{i\in[M]}$, where $[M] = \{1, \ldots, M\}$. Each bandit instance $\nu_i$, which we will sometimes call a *task*, is a *linear contextual bandit* (Wang et al., 2005) that maps an action $k \in [K]$ and context $x \in \mathcal{X} \subseteq \mathbb{R}^d$ into a reward distribution $\nu_i(x, k) = x^\top \theta_{ik} + \eta_{ik}$, where $\theta_{ik} \in \mathbb{R}^d$ is a vector of parameters and $\eta_{ik}$ is a (subgaussian) random noise with zero mean and variance $\sigma_{ik}^2 \leq \sigma^2$. A special yet important case is when the space of contexts is a singleton $\mathcal{X} = \{x\}$, which we call *non-contextual* bandit, or just bandit for simplicity.

Following a typical *stochastic* bandit setup (Lattimore & Szepesvári, 2020), the decision maker, i.e., the *agent*, interacts with a bandit $\nu_i \in \mathbb{M}$, which identity is not revealed to the agent. The interaction protocol goes as follows: At each step $t > 0$, the agent observes a context $x_t \in \mathcal{X}$ drawn from some fixed distribution $\mathcal{P}$, it selects an arm $k_t \in [K]$, and it collects a reward $r_t \sim \nu_i(x_t, k_t)$. The agents decides the arm to pull according to a policy $\pi : \mathcal{X} \to [K]$, a mapping between contexts and arms, which the agent updates given previous observations of contexts and rewards.

The goal of the agent is to maximize the cumulative reward collected over a time horizon $H$ or, equivalently, to minimize the *regret* of pulling an arm other than the optimal one. For instance, to minimize the number of times a treatment different from the optimal one is administered to a patient. Since the identity of the bandit problem (unobserved charac-

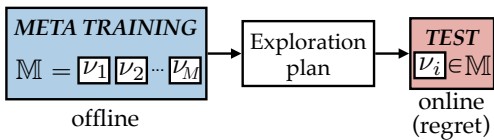

Figure 2: The meta learning bandits problem setting.

teristic of the patient in the example) is hidden to the agent, the regret is typically computed over the worst-case task in $\mathbb{M}$. Formally, the *worst-case* regret is given by

$$\text{Reg}_H(\mathbb{M}) := \sup_{\nu_i \in \mathbb{M}} \mathbb{E}\left[\sum_{t \in [H]} \max_{k \in [K]} x_t^\top \theta_{ik} - r_t\right] \quad (1)$$

where the contexts $x_1, \ldots x_H$ are sampled independently from the fixed distribution $\mathcal{P}$ and $r_t \sim \nu_i(x_t, k_t)$ being $k_t \sim \pi(x_t)$ the arm pulled by the agent.

In this paper, we consider a *meta learning* variation (*e.g.*, Cella et al. 2020; Kveton et al. 2020) of the common bandit setup described above. The learning setting (Figure 2) is composed of two separate and consecutive stages, which we call *meta training* and *test*, respectively.

**Meta training.** In the first stage, the agent can interact *offline* with the set of bandits $\mathbb{M}$. Differently from a *pure exploration* setup (Audibert & Bubeck, 2010), here we interact with a set of bandits instead of a single one. We are not just interested in discovering an optimal policy for each bandit, but also to devise an *exploration plan*, which we denote as $\text{Plan}(\mathbb{M})$, that we can transfer to the test phase to minimize the regret. Since the meta training itself happens entirely offline, no regret is incurred at this stage. In practice, this is reasonable when working with a simulator or previously collected data, such as an historical record of treatments administered to patients. However, we may operate under resource constraints, so that it is important to investigate the sample and computational complexity of meta training.

**Test.** In the second stage, the agent faces a single and unknown bandit task $\nu_i \in \mathbb{M}$, which we call the *test task*, with the goal of minimizing the regret (1). This matches the *stochastic* bandit setting exactly, except that the learning algorithm takes decisions according to the exploration plan devised during meta training, *i.e.*, $k_t \sim \text{Plan}(\mathbb{M})$. Whereas the plan is fixed *a priori*, it is still *adaptive*, as it conditions the decisions with the history of interactions in the test task. For instance, the plan can be a strategy to administer treatments to a patient informed by historical data.

What are the theoretical barriers for the described problem of meta learning bandits? A natural question is whether the meta training can benefit the test regret in a substantial way. Perhaps unsurprisingly,[1] without any assumption on how

the collection of bandits $\mathbb{M}$ is constructed, the meta learning problem is not easier than the classical stochastic bandit.

**Theorem 2.1** (Lai & Robbins 1985). *Let $\mathbb{M}$ a set of $M \geq 2$ bandits and let $\mathcal{X} = \{x\}$ be a singleton. The test regret is $\text{Reg}_H(\mathbb{M}) = \Omega(\sqrt{MH})$.*

The latter can be proved through a hard instance in which the two bandits are identical expect for a pair of arms whose mean reward differ for a small quantity depending on $H$. In many scenarios, those instances have limited interest, as we may model the pair of bandits with a single task, at the cost of a (bounded) sub-optimality. Similarly to previous meta learning settings (Chen et al., 2022b; Mutti & Tamar, 2024), we consider a *separation* assumption built on this premise.

**Assumption 1.** *For all $i \neq j \in [M]$ and a policy class $\Pi$, there exists at least one policy $\pi \in \Pi$, s.t. $D_H(\mathbb{P}_i^\pi, \mathbb{P}_j^\pi) \geq \lambda$, where $D_H$ is the Hellinger distance and $\mathbb{P}_i^\pi, \mathbb{P}_j^\pi$ are the joint context-arm-reward distributions induced by $\pi$ in $\nu_i, \nu_j$.*

The separation guarantees that the bandits in the collection are meaningfully different, such as assuming that different patient groups respond differently to at least one treatment.

We have now a formal picture of the setting we consider: Meta learning bandits under separation. Before going ahead with the investigation of the setting, we introduce additional notation for later use.

**Notation.** We will consider a fixed context distribution $\mathcal{P}$ for both meta training and test stages. For a random variable $A$ and event $\mathcal{E}$, we use $\mathbb{E}_\mathcal{P}[A], \mathbb{P}_\mathcal{P}[\mathcal{E}]$ as shortcuts for $\int_{x \in \mathcal{X}} \mathcal{P}(x) \mathbb{E}[A|x] dx$ and $\int_{x \in \mathcal{X}} \mathcal{P}(x) \mathbb{P}(\mathcal{E}|x) dx$ respectively. For any finite set $S$, we denote $2^S$ the powerset of $S$. For any two probability distributions $p, q$ over some measurable space $\mathcal{X}$, let $D_H(p, q) := \int_{x \in \mathcal{X}} \left(\sqrt{p(x)} - \sqrt{q(x)}\right)^2 dx$ be the Hellinger distance between them. For every $\nu_i \in \mathbb{M}$, we denote $\mu_{ik} = \mathbb{E}_\mathcal{P}[x^\top \theta_{ik}]$ the mean of $r \sim \nu_i(x, k)$ for $x \sim \mathcal{P}$. We further assume $x^\top \theta_{ik} \in [0, 1]$ and both $\|x\|_1, \|\theta_{ik}\|_1$ to be bounded. We denote as $\Pi$ the space of policies and the *optimal policy* $\pi_i^*(x) := \arg\max_{\pi \in \Pi} x^\top \theta_{i\pi(x)}$, playing the arm $k_i^* \in \arg\max_{k \in [K]} x^\top \theta_{ik}$ with the optimal mean reward for any $x \in \mathcal{X}$. For a bandit $\nu_i \in \mathbb{M}$ and policy $\pi \in \Pi$, we denote $\mathbb{P}_i^\pi$ the joint distribution of context-arm-rewards. The *action gap* of bandit $\nu_i$ and context $x$ is denoted $\Delta_i(x, k) := x^\top \theta_{ik^*} - x^\top \theta_{ik}$ and we define $\Delta := \min_{i \in [M], x \in \mathcal{X}, k \in [K]} \Delta_i(x, k)$.[2]

---

[1]The result is obvious from classical minimax lower bound

constructions for stochastic bandits. See the one in Chapter 15 of Lattimore & Szepesvári (2020) for a gentle introduction.

[2]Note that, whenever the context vector is the zero vector, the gap $\Delta_i$ collapses to zero for every $i$. We assume that the space of contexts $\mathcal{X}$ is designed properly, so that it does note include such dummy context vectors.

## 3. Meta learning bandits with classification

In this section, we present a framework to study meta learning bandits under separation through the lenses of multiclass classification. First, we analyze the regret of a strategy, *i.e.*, an exploration plan Plan($\mathbb{M}$), based on classifying the test task to then exploit the optimal policy of the classified task. Then, we show that classifying the test is necessary for regret minimization under separation. As we shall see, the two results are brought together by a novel measure of complexity, which we call the *classification-coefficient*.

For the ease of presentation, we assume to know the true distributions of all bandits $\nu_i \in \mathbb{M}$, and we leave the study of misspecifications to later sections. We consider classification algorithms in the following interaction protocol:

1. Start with $t = 0$ and an initial hypothesis class $S_0 = \{1, 2, ..., M\}$.
2. Terminate if $|S_t| = 1$. Otherwise, decide on a classification test $\pi_t \in \Pi_\mathcal{C}$ (either deterministically or randomly) from the set of tests $\Pi_\mathcal{C}$, and draw $N_\text{cls} = \tilde{O}(\lambda^{-2})$ samples with $\pi_t$.
3. Update the hypothesis class $S_{t+1}$ with the generated samples. $t \leftarrow t + 1$ and go to Step 2.

The complexity of classification depends on how many hypotheses we can rule out from a test $\pi_t$ from the remaining hypotheses each round. As we are allowed to use $\tilde{O}(\lambda^{-2})$ samples, we can at least rule out $\lambda$-separated hypotheses from the underlying instance. Specifically, given the remaining hypothesis class $S_t \in 2^{[M]}$ and the underlying instance $i$, we can remove $\bar{S}_{t,\lambda}^\pi(i) := \{m \in S_t | D_\text{H}(\mathbb{P}_i^\pi, \mathbb{P}_m^\pi) \geq \lambda\}$ through hypothesis testing (*e.g.,* using likelihood ratio test).

To formalize the concept, we define the deterministic *classification-coefficient*:

$$C_\lambda(\Pi_\mathcal{C}) := \max_{S \in 2^{[M]}, |S| > 1} \min_{\pi \in \Pi_\mathcal{C}} \max_{i \in S} \frac{|S|}{|\bar{S}_\lambda^\pi(i)|}, \quad (2)$$

and the randomized *classification-coefficient*:

$$\widetilde{C}_\lambda(\Pi_\mathcal{C}) := \max_{S \in 2^{[M]}, |S| > 1} \min_{p \in \Delta(\Pi_\mathcal{C})} \max_{i \in S} \frac{|S|}{\mathbb{E}_{\pi \sim p}[|\bar{S}_\lambda^\pi(i)|]}, \quad (3)$$

In essence, these coefficients measure the classification complexity of a class of bandits through the pessimistic rounds of classification, where $S$ is the worst-case remaining hypotheses when the test task is $i$, and $\pi, p$ are the optimal deterministic and randomized greedy strategies, respectively. The latter take the test (resp. distribution over tests) inducing the most even split (resp. expected split) of the remaining hypotheses $S$. Interestingly, we can derive an upper bound on the size of the split when employing the deterministic greedy strategy

$$\mathbb{E}\left[\frac{|S_{t+1}|}{|S_t|}\Big| S_t\right] \leq 1 - \frac{1}{2}C_\lambda(\Pi_\mathcal{C})^{-1}.$$

**Algorithm 1** Explicit Classify then Exploit

1: **input** set of tasks $\mathbb{M}$, $N_\text{cls}$
2: Initialize $S_0 = [M], t = 0$      Explicit Classify
3: **while** $|S_t| > 1$ **do**
4:    $\pi_t = \max_{\pi \in \Pi_\mathcal{C}} \min_{i \in S_t} |\bar{S}_{t,\lambda}^\pi(i)|$
5:    $\mathcal{D}_t \leftarrow N_\text{cls}$ i.i.d. samples drawn with $\pi_t$
6:    Get $S_{t+1}$ with Algorithm 2
7:    $t \leftarrow t + 1$
8: **end while**
9: Extract the classified task $m^* \in S_t$ and execute $\pi^*(x) = \arg\max_{\pi \in \Pi} \nu_{m^*}(x, k)$ for the remaining steps    Exploit

---

**Algorithm 2** Update Remaining Hypotheses

1: **input** set of tasks $S_t$, test $\pi_t$, samples $\mathcal{D}_t$
2: Let $\ell_i = \sum_{(x,r) \in \mathcal{D}_t} \log(\mathbb{P}_i^{\pi_t}(x, r))$ for all $i \in S_t$
3: Let $\hat{m} = \arg\max_{i \in S_t} \ell_i$
4: **return** $S_{t+1} \leftarrow \{i \in S_t | \ell_i \geq \ell_{\hat{m}} - 3\log(M/\delta)\}$

---

Clearly, the smaller the classification-coefficients, the more hypothesis we can rule out in a single round, the easier it is to classify the test task. In the following result, we formally link the complexity of classification with the regret.

To this end, we consider a simple algorithm, called *Explicit Classify then Exploit* (ECE, Algorithm 1), which is based on the classification protocol described above to classify the test task (lines 2-8), then deploying the optimal policy for the classified task (line 9). We can prove the following.

**Theorem 3.1.** *Suppose Assumption 1 holds with a test class $\Pi_\mathcal{C}$ and a family of $M$ bandit instances $\mathbb{M}$. Then with probability at least $1 - \delta$, the while-loop in Algorithm 1 ends after $T$ rounds with $N_\text{cls}$ samples per round where*

$$T = \mathcal{O}\left(C_\lambda(\Pi_\mathcal{C})\log(M/\delta)\right),$$
$$N_\text{cls} = \mathcal{O}\left(\log(M/\delta)/\lambda^2\right). \quad (4)$$

*Consequently, the expected test regret of Algorithm 1 for $H$ steps is*

$$\text{Reg}_H(\mathbb{M}) \leq \mathcal{O}\left(\frac{C_\lambda(\Pi_\mathcal{C})\log^2(M/\delta)}{\lambda^2}\right) + \delta H.$$

The theorem states that we can identify the test task w.h.p. taking $N_\text{cls}T = O(\lambda^{-2} \cdot C_\lambda(\Pi_\mathcal{C})\log^2(M/\delta))$ samples. We can translate the latter into a regret rate by bounding the regret caused by classification failure with $\delta H$. We can set $\delta = o(1/H)$ to make the classification failure negligible, settling the regret $O(\lambda^{-2}C_\lambda(\Pi_\mathcal{C})\log^2(MH))$.[3] Next, we show that the latter rate is nearly optimal by developing a lower bound to the regret for bandits under separation.

---

[3]For randomized classification, we can change Algorithm 1 to perform a randomized test, and the same conclusion holds with replacing $C_\lambda$ by $\widetilde{C}_\lambda$.

## 3.1. Necessity of classification with separation

While the ECE approach may not always be the best algorithm to minimize regret, it is a near-optimal solution whenever the optimal actions and the separating actions do not overlap. To see this, suppose a family of $M$ multi-armed bandit instances $\mathbb{M}$ with arbitrarily many $K$ arms. Each $i^{th}$ instance has its unique optimal arm $k_i^*$, but only with margin $O(\epsilon)$, *i.e.*, instances are not well-separated with respect to optimal arms. In such scenarios, it is always better to first identify the task with $\lambda$-separating arms.

To formalize the fundamental link between regret and classification, for the remainder of the section we are going to consider a class of worst-case multi-armed bandit instances $\mathbb{M}$, which we refer as `hard`, defined as follows:

1. For each bandit instance $i \in [M]$, there is a unique optimal arm $k_i^* \in [K]$ such that

$$\mu_i(k_i^*) = \frac{3}{4} + 10\epsilon, \ \mu_j(k_i^*) = \frac{3}{4}, \ \forall j \neq i.$$

2. All other arms $k \in [K]/\{k_i^*\}_{i \in [M]}$ are information-revealing, *i.e.*, either one of the following holds:

$$\mu_i(k) = \frac{1+\lambda}{2} \text{ or } \mu_i(k) = \frac{1-\lambda}{2}, \ \forall i \in [M],$$

where $\epsilon, \lambda$ satisfy $1 > \lambda^2 > c_\lambda \epsilon \cdot \widetilde{C}(\mathbb{M})$ for some sufficiently large absolute constant $c_\lambda > 0$ and the randomized classification-coefficient $\widetilde{C}(\mathbb{M})$ (defined below).

**Classification complexity.** Let $C^*(\mathbb{M})$ be the *optimal* depth of a deterministic decision tree classifier for the `hard` instance, constructed by probing the true means of separating arms $\mathcal{A}_\lambda := [K]/\{k_i^*\}_{i \in [M]}$. Let $\widetilde{C}^*(\mathbb{M})$ be the optimal average depth of randomized decision trees. In this case, the classification-coefficient in (2) can be defined as $C(\mathbb{M}) := C_\lambda(\mathcal{A}_\lambda)$, and similarly for the *randomized* classification-coefficient $\widetilde{C}(\mathbb{M}) := \widetilde{C}_\lambda(\mathcal{A}_\lambda)$.[4] Note that the *classification-coefficients* defined previously are concerned with the (worst-case) most even split on the hypotheses $S_t$, and thus they can be interpreted as measures for greedy classification strategies. The following is a well-known relationship between these greedy measures and the optimal depth of (deterministic) decision trees (Arkin et al., 1993)

$$\widetilde{C}(\mathbb{M}) \leq C(\mathbb{M}) \leq C^*(\mathbb{M}) \leq C(\mathbb{M}) \log(M). \quad (5)$$

We note that these classification complexities can be as large as $M$ in the worst case, while in practical scenarios we can often design effective information-revealing actions to ensure $C^*(\mathbb{M}) = O(\log M)$.

**Statistical barriers of separated bandits.** What is the lower bound to the test regret for $\mathbb{M}$? To quantify this,

we recall a PAC-variant of DEC from (Chen et al., 2022a). Specifically, given some $\gamma > 0$, we define the coefficient

$$\mathtt{dec}_\gamma(\mathbb{M}) := \max_{\omega \in \Delta([M])} \min_{\pi \in \Delta([K])} \max_{i \in [M]}$$

$$\mathbb{E}_{k \sim \pi}[\Delta_i(k)] - \gamma \mathbb{E}_{k \sim \pi, m \sim \omega}[D_H^2(\nu_i(k), \nu_m(k))], \quad (6)$$

where $\Delta_i(k) := \mu_i(k_i^*) - \mu_i(k)$. We can verify the following relation between $\gamma$ and $\mathtt{dec}_\gamma$:

**Lemma 3.2.** *There exists an absolute constant $c_\gamma > 0$ such that for all $\gamma \leq c_\gamma \lambda^{-2} \widetilde{C}(\mathbb{M})$, we have $\mathtt{dec}_\gamma(\mathbb{M}) > 3\epsilon$.*

As a corollary of (Chen et al., 2022a, Theorem 10), this implies the lower bound on the high probability regret:

**Theorem 3.3.** *There exists an absolute constant $c > 0$, such that if $1/H < c\epsilon$, then any algorithm must suffer regret $\Omega(\min(\epsilon H, c_\gamma \lambda^{-2} \widetilde{C}(\mathbb{M})))$ with probability at least $1/H$.*

Thus, any algorithm guarantees with probability at least $1 - 1/H$ must suffer at least $\Omega(\widetilde{C}(\mathbb{M})\lambda^{-2})$ test *regret*, capturing the fundamental limits of separated bandits. Note that the lower bound depends on the *randomized* classification-coefficient, though deterministic strategies can still be preferred due to their simplicity in practice.

# 4. A more practical ECE algorithm

In the previous section, we analyzed the ECE algorithm in an *ideal* setting in which the reward distributions of all the bandits in $\mathbb{M}$ and the context distribution $\mathcal{P}$ are fully known.[5] Here, we present a more practical variation of the algorithm, *Decision Tree ECE* (DT-ECE), which (i) is robust to misspecifications of $\mathbb{M}$ caused by estimation errors at meta training, (ii) only accesses samples coming from the context distribution $\mathcal{P}$, (iii) lays down a fully interpretable exploration plan through a decision tree classifier.

In this section, we work under a special case of the separation condition (Ass. 1) which assumes separation on the mean of the rewards instead of their distribution.

**Assumption 2.** *For some $\lambda > 0$ and every $\nu_i, \nu_j \in \mathbb{M}$, there exists $k \in [K]$ such that $|\mathbb{E}_{x \sim \mathcal{P}}[x^\top(\theta_{ik} - \theta_{jk})]| > \lambda$.*

First, we describe the meta training stage with the corresponding estimation guarantees, sample and computational complexity (Section 4.1). Then, we present the DT-ECE test algorithm and we analyze its regret (Section 4.2).

## 4.1. Meta training

In this section, we describe a provably efficient algorithm to meta train an exploration plan `Plan(M)` by only accessing offline simulators of the tasks in $\mathbb{M}$ and samples from $\mathcal{P}$.[6]

---

[4] From here on, we denote a *classification-coefficient* $C_\lambda(\Pi_{\mathcal{C}})$ as $C_\lambda(\mathbb{M})$ when the space of tests $\Pi_{\mathcal{C}}$ is a (sub)set of arms.

[5] Algorithm 2 access $\mathcal{P}$ to compute the log likelihood at step 2.

[6] An analogous algorithm accessing pre-logged historical data can be developed. The reported guarantees shall transfer verbatim

**Algorithm 3** Meta Training

1: **input** simulators $\mathbb{M}$, $N_{\text{est}}$
2: Initialize $\hat{\mathbb{M}} = \emptyset$
3: **for** $i \in [M]$ **do**
4:    **for** $k \in [K]$ **do**
5:       Sample $N_{\text{est}}$ contexts $X = (x_n \sim \mathcal{P})$
6:       Sample $N_{\text{est}}$ rewards $\boldsymbol{r} = (r_n \sim \nu_i(x_n, k))$
7:       Compute $\hat{\theta}_{ik} = (XX^\top)^{-1}X\boldsymbol{r}$
8:       Compute $\hat{\mu}_{ik} = \frac{1}{N_{\text{est}}} \sum_n r_n$
9:    **end for**
10:   $\hat{\mathbb{M}}$.append($\hat{\nu}_i = ([\hat{\theta}_{i1}, \hat{\mu}_{i1}], \dots [\hat{\theta}_{iK}, \hat{\mu}_{iK}])$)
11: **end for**
12: Build a decision tree classifier $\texttt{tree}(\hat{\mathbb{M}})$ with Algorithm 4
13: **output** exploration plan $\texttt{Plan}(\hat{\mathbb{M}})$ prescribed by $\texttt{tree}(\hat{\mathbb{M}})$

---

The meta training algorithm, whose pseudocode is in Algorithm 3, has two main procedures. First, it estimates the parameters of each task $\nu_i$ by doing regression on the class of linear functions of the context (lines 2-11). Second, it takes the (possibly misspecified) resulting class $\hat{\mathbb{M}}$ to build a deterministic decision tree classification model over the tasks (line 12). The following lemma provides an estimation guarantee over $\hat{\mathbb{M}}$ from the analysis of *random design* linear regression (Hsu et al., 2011).

**Lemma 4.1.** *Let $\mathbb{M}$ be a set of $M$ linear contextual bandits and let $\hat{\mathbb{M}}$ their estimation obtained by Algorithm 3 with*

$$N_{\text{est}} = \frac{160\sigma^2 d \log(4HMK)}{\min(\Delta^2, \lambda^2)}.$$

*For every bandit $i \in [M]$ and arm $k \in [K]$, it holds*

$$\mathbb{P}\left(\mathbb{E}_{\mathcal{P}}\left[|x^\top \hat{\theta}_{ik} - x^\top \theta_{ik}|\right] > \min\left(\frac{\Delta}{2}, \frac{\lambda}{4}\right)\right) \leq \frac{1}{2HMK}.$$

The latter guarantees that the identity of the optimal arm and the separation condition is preserved w.h.p. by the estimation process. As we shall see, these properties will prove useful at test stage. Before going to that, it is worth detailing how the decision tree classifier is built (Algorithm 4).

We consider a set of tests $\Pi_{\mathcal{C}}$ equal to the set of arms $[K]$, for which we are going to test the mean reward $\hat{\mu}_k$ against a threshold $b \in [0, 1]$. Since computing the optimal test is NP-hard in general (Hyafil & Rivest, 1976), we turn to a greedy approximation which gives the test with the most even split (Arkin et al., 1993; Nowak, 2011). Algorithm 5 in Apx. C.1 gives a tractable procedure with which the greedy test can be computed. In order to make the tests along the tree statistically robust when computed with samples from the test task, we consider *soft splits* (Olaru & Wehenkel, 2003): We let the test $\hat{\mu}_k \leq b$ be simultaneously true and false inside a $\lambda$-band around $b$ (see Figure 3).

---

under natural conditions on the size and quality of the dataset.

**Algorithm 4** Decision Tree

1: **input** set of tasks $S$
2: **if** $|S| > 1$ **then**
3:    Compute $(\mu_k \leq b) \leftarrow \texttt{greedy}(S)$ with Algorithm 5
4:    Define $\texttt{tree}(S) := (\mu_k \leq b)$
5:    Compute $S^+ = \{\nu_i \in S \mid \mu_{ik} \leq b + \lambda/2\}$
6:    Compute $S^- = \{\nu_i \in S \mid \mu_{ik} > b - \lambda/2\}$
7:    Define $\texttt{tree}(S, \text{true}) := S^+$ and $\texttt{tree}(S, \text{false}) := S^-$
8:    Call Algorithm 4 on $S^+$ and $S^-$ recursively
9: **end if**

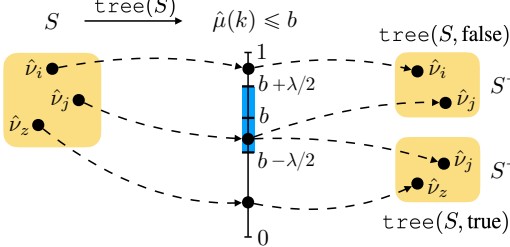

Figure 3: Visualization of a generic split of $\texttt{tree}(\hat{\mathbb{M}})$.

The meta training algorithm that we just described is *fully tractable*, both in terms of computational resources and sample complexity, as proved by the result below.

**Theorem 4.2.** *Algorithm 3 runs in time $\mathcal{O}(d^3 M^3 K/\lambda^4)$ and collects a total number of samples*

$$\frac{160\sigma^2 MKd \log(4TMK)}{\min(\Delta^2, \lambda^2)}.$$

Finally, we can provide a guarantee on the cost of the greedy approximation with respect to the depth of the optimal deterministic decision tree on $\hat{\mathbb{M}}$, *i.e.*, $C_\lambda^*(\hat{\mathbb{M}})$.

**Lemma 4.3.** *Algorithm 4 builds a decision tree with depth $D = \mathcal{O}(\log M + 1)C_\lambda^*(\hat{\mathbb{M}})$.*

### 4.2. Test

Here we analyze the test algorithm implementing the exploration plan $\texttt{Plan}(\hat{\mathbb{M}})$ prescribed by the decision tree classifier $\texttt{tree}(\hat{\mathbb{M}})$, which we call DT-ECE. As said above, this test algorithm is a slight variation of ECE (Algorithm 1) and mostly follow similar steps. Here we comment on the differences and we leave a complete pseudocode to Apx. C.2.

Without turning to the appendix, we can look at the pseudocode in Algorithm 1 and picture that, at line 4, DT-ECE would extract a test $\mu_k \leq b$ from $\texttt{tree}(S_t)$ on the current hypotheses $S_t$, collecting data like in line 5 with the policy $\pi_t = k$ prescribed by the test. Then, instead of updating the remaining hypotheses $S_{t+1}$ with log likelihood tests (line 6), it takes $S_{t+1}$ by following the left or right split in the tree according to whether the test resulted true or false, respectively. Those changes lead to the following regret.

**Theorem 4.4.** *Suppose Assumption 2 holds on a set of tasks $\mathbb{M}$ and let* $\texttt{tree}(\hat{\mathbb{M}})$ *be obtained from Algorithm 3. The expected test regret of DT-ECE (Algorithm 6) for $H$ steps is*

$$\text{Reg}_H(\mathbb{M}) = \mathcal{O}\left(\frac{C_\lambda^*(\mathbb{M})\log^2(C_\lambda^*(\mathbb{M})MH)}{\lambda^2}\right)$$

The result above shows that DT-ECE matches the regret of ECE with a factor $C_\lambda^*(\mathbb{M})$ in place of the *classification-coefficient* $C_\lambda(\mathbb{M})$. This implies an additional $\log(M)$ factor at most (see 5). This means the estimation error does not significantly affect the regret, thanks to the guarantee in Lemma 4.1. Finally, the regret holds in a contextual bandit setting, but does not depend on the size of the context $d$, which only impacts the meta training complexity.

## 5. Experiments

In this section, we provide a brief numerical validation to illustrate how the above theoretical analysis on the classification view of meta learning bandits translates to compelling empirical results, which we compare with previous methods in the literature of latent bandits (Hong et al., 2020a).[7]

To the purpose of the experiments, we consider a non-contextual stochastic MAB setting in which the collection of bandits is fully known, without covering class misspecifications. We design two family of collections, one inspired by the hard instance presented in Section 3.1, which we henceforth call `hard`, and one randomly generated collection, which we call `rand`. For the former, we consider two instances with size $M = 5$ and arms $K = 10$, with varying values of the separation parameters $\lambda$ (0.4 and 0.04 respectively). For the latter, we consider a small instance $M = 10, K = 20$ and a large instance $M = 40, K = 40$. We use rejection sampling to control $\lambda$ (set to 0.4) in the randomly generated collection. In all the considered instances, the reward distributions are Bernoulli.

We compare the regret suffered by our decision tree implementation of the *Explicit Classify then Exploit* routine (DT-ECE, described in Section 4 and Algorithm 6 of Apx. C.2) with traditional bandit approaches, *i.e.*, mUCB (Azar et al., 2013) and mTS (Hong et al., 2020a). The latter algorithms adapt UCB and Thompson sampling to the meta/latent bandits setting. While they are not designed to take advantage of separation specifically, they exploit knowledge of the collection of bandits and they constitute relatively strong baselines. Before going ahead with the experimental results, it is worth spending a few words on how the *spirit* of our algorithm differs to theirs. DT-ECE is designed to produce easy-to-interpret exploration plans, which can be entirely pre-computed offline. Instead, the exploration prescribed

---

by mUCB and mTS is hardly interpretable nor predictable, making them and DT-ECE orthogonal solutions for different applications rather than direct challengers. It is satisfying, however, to see that DT-ECE performance is on par with such renowned algorithms.

In Figure 4 (a, b) we see that DT-ECE achieves a small regret by classifying the test task in a handful of interactions (coarsely, the classification occurs at the elbow of the curves) both when separation is large (a) or small (b). DT-ECE is able to commit to the optimal strategy even before mTS, whose posterior takes slightly longer to converge around the test task, although DT-ECE suffers larger regret due to pure exploration. The most important trait of the `hard` instance is that optimal actions and informative actions do not overlap, so that optimistic strategy like mUCB are bound to fail. By mostly pulling nearly optimal yet non-informative actions, mUCB cannot identify the test task efficiently, and the regret grows steady. Optimism works considerably better in the `rand` family (Figure 4 c, d), although mUCB does not match the efficiency of DT-ECE and mTS in those experiments either. It is remarkable that DT-ECE can classify the test task into a set of 40, with 40 arms each, by taking less than 1000 samples on average (d).

Finally, DT-ECE comes with sharp theoretical guarantees and it is designed for the worst case, which can limit the performance of the algorithm in more forgiving instances (such as the `rand` family). However, the design of a fully practical version of the ECE ideas is beyond the scope of this paper and constitute interesting matter for future studies.

## 6. Related work

To the best of our knowledge, our classification view of meta learning bandits under separation is original and has not been studied. There are anyway several connections between our results and the literature, which we revise below.

**Contextual bandits.** Obviously, our setting relates to *contextual* bandits (Wang et al., 2005; Li et al., 2010; Abbasi-Yadkori et al., 2011; Hao et al., 2020) and, indeed, our results hold for the contextual setting. The contextual nature of individual tasks is an orthogonal dimension w.r.t. a second, *unobserved* context typical of meta learning settings: The task description itself.

**Latent bandits.** The setting that most closely relates to ours is *latent bandits* (Azar et al., 2013; Maillard & Mannor, 2014; Zhou & Brunskill, 2016; Hong et al., 2020a;b; Pal et al., 2023). Actually, our setting can be seen as a particular instance of latent bandits under separation and a meta learning protocol. Azar et al. (2013); Maillard & Mannor (2014) also consider bandit tasks coming from a finite and known set, with or without misspecification. They do not consider separation, which allows to specialize the regret

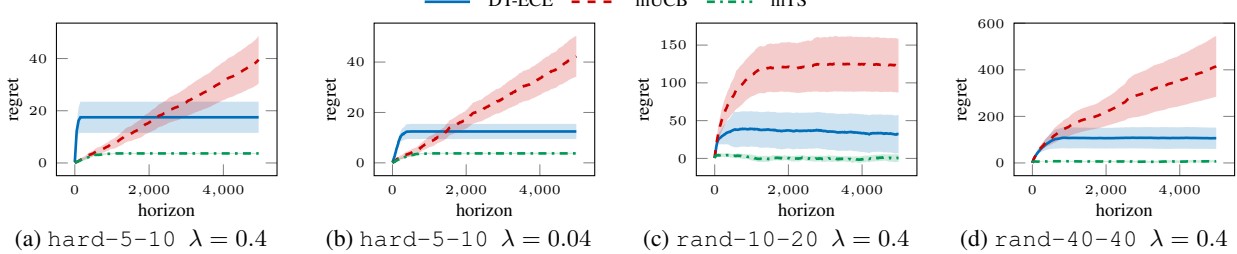

Figure 4: Regret of DT-ECE (ours), mUCB (Azar et al., 2013), mTS (Hong et al., 2020a). Captions report `envname-M-K`, denoting the name of the collection of bandits, the size of the collection, and the number of arms, respectively, together with the value of the separation parameter $\lambda$. The curves average 20 independent runs, shaded regions are 95% c.i.

from $\mathcal{O}(\sqrt{H})$ to $\mathcal{O}(\log H)$. Similarly to ours, the setting in (Zhou & Brunskill, 2016) includes a phase in which the models are learned from data and then exploited on future tasks. In their formulation, however, the tasks are coming into a sequence online, so that the meta learning itself adds to the regret instead of being carried out offline. An offline learning phase is considered by Hong et al. (2020a) in a problem formulation that almost perfectly matches ours, yet leads to mostly orthogonal results: They do not consider separation; Their analysis is not instance-dependent and does not tie the regret to the classification complexity of the instance; They consider traditional UCB/TS-style algorithms in place of our ECE; They do not detail the meta training algorithm. Most importantly, our classification view is original in the latent bandits literature and constitutes the main novelty of our work.

**Low-rank bandits.** *Low-rank bandits* (Kveton et al., 2017; Lale et al., 2019; Lu et al., 2021) essentially generalize the latent bandits formulation (and ours) by assuming the existence of a low-rank latent representation conditioning the arms payoffs. Just like in latent bandits, previous works do not touch on the connection between classification and regret, which may be generalized to low-rank bandits.

**Structured bandits.** In *structured bandits* (Lattimore & Munos, 2014; Combes et al., 2017; Tirinzoni et al., 2020) the rewards of the arms are correlated according to a known structure *class* with hidden parameters. These parameters have some similarity of the hidden task context of our setting (and latent bandits). Our results connecting classification and regret may be generalized to structured bandits.

**Thompson sampling.** Extensive work has been done over exploiting prior knowledge in bandits through Bayesian approaches. The most notable is Thompson sampling (Thompson, 1933; Kaufmann et al., 2012; Agrawal & Goyal, 2012; Russo & Van Roy, 2016), in which knowledge over the test task is incorporated into a prior. The set of tasks of our setting can be seen as a prior, although our results are in a frequentist setting. As such, they are independent from the prior distribution and robust to misspecifications, differently

from Thompson sampling (Simchowitz et al., 2021).

**Meta learning bandits.** Meta learning bandits has been considered in (Kveton et al., 2021; Hong et al., 2022b;a) where tasks are assumed to come from an unknown prior. The agent aims to infer the prior from interaction, assuming it is itself coming from a known hyper-prior. This can be seen as a Bayesian version of our setting, where the hyper-prior stands for the set of tasks, and the priors play the role of the tasks. Related to this stream, other works (Cella et al., 2020; Basu et al., 2021) have considered meta learning a prior over tasks for regret minimization.

## 7. Conclusion

In this paper, we took an original *classification view* on the problem of meta learning bandits under separation. Thanks to this novel approach, our work delivers on its promise of providing principled algorithms for learning *interpretable* and *efficient* exploration plans from offline data, just like they were designed by humans. As a by product to this effort, we contribute an elegant *framework* to study the regret of learning algorithms through the complexity of classifying the task *online* within a set of previously seen tasks.

We believe the significance of our findings are hardly limited to the considered contextual multi-armed bandits, and that they may inspire future works targeting yet more general problem settings (and corresponding applications) by following our blueprint for meta learning with classification.

A natural next step is to introduce dynamics over contexts to extend the framework to full-fledged Markov Decision Processes (MDPs) and reinforcement learning, where we would consider a test MDP coming from a collection of MDPs, known a priori or accessed offline. A framework of similar kind has been introduced under the name of *contextual* MDPs (Hallak et al., 2015) and latent MDPs (Kwon et al., 2021b;a; 2023b;a; 2024). Previous works have also studied meta learning policies for efficient exploration in MDPs and their regret (Chen et al., 2022b; Ye et al., 2023; Mutti & Tamar, 2024). None of the above has considered

our classification view of the problem to get efficient and interpretable exploration plans. In the MDP setting, our decision tree classifier resembles a hierarchical strategy deploying policies, or *options* (Sutton et al., 1999), to probe information-revealing states of the environment. Can these policies be learned with a tractable offline algorithm? Would the exploration plan enjoy similar regret guarantees beyond the contextual MAB setting? This is an exciting direction with the potential to open the door to countless applications, such as autonomous driving, robotics, and many others.

## Acknowledgements

This research was partly funded by the European Union (ERC, Bayes-RL, 101041250). Views and opinions expressed are however those of the author(s) only and do not necessarily reflect those of the European Union or the European Research Council Executive Agency (ERCEA). Neither the European Union nor the granting authority can be held responsible for them.

## Impact statement

This paper presents work whose goal is to advance the field of Machine Learning. There are many potential societal consequences of our work, none which we feel must be specifically highlighted here.

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

## A. Auxiliary Lemmas

The following lemma is the famous Ville's inequality for super-martingales:

**Lemma A.1** (Ville's Inequality). *Let $\{W_t\}_{t\geq 0}$ be a non-negative super-martingale sequence, such that*

$$\mathbb{E}[W_{t+1}|W_t] \leq W_t,$$

*for any $\delta > 0$, the following holds:*

$$\mathbb{P}\left(\forall t, W_t \leq W_0/\delta\right) \geq 1 - \delta.$$

The following lemmas are the standard concentration of log-likelihood values of the models within the confidence set. The proofs are standard in model-based RL and can also be found in (*e.g.*, Liu et al. 2022; Agarwal et al. 2020). We let $\mathcal{D}$ be the observational data $o = (x, k, r)$ collected by running $\pi$ on some underlying distribution $\nu^* \in \mathbb{M}$. We denote $\beta := \log(M/\delta)$. Then, the following holds:

**Lemma A.2** (Uniform Bound on the Likelihood Ratios). *With probability $1 - \delta$ for any $\delta > 0$, for any $\nu \in \mathbb{M}$,*

$$\sum_{o\in\mathcal{D}} \log(\mathbb{P}_\nu^\pi(o)) - \beta \leq \sum_{o\in\mathcal{D}} \log(\mathbb{P}_{\nu^*}^\pi(o)). \tag{7}$$

**Lemma A.3** (Concentration of Maximum Likelihood Estimators). *With probability $1 - \delta$, for all $\nu \in \mathbb{M}$, we have*

$$D_{\mathrm{H}}^2(\mathbb{P}_\nu^\pi, \mathbb{P}_{\nu^*}^\pi) \leq \frac{1}{2|\mathcal{D}|}\left(\sum_{o\in\mathcal{D}} \log\left(\frac{\mathbb{P}_{\nu^*}^\pi(o)}{\mathbb{P}_\nu^\pi(o)}\right) + 3\beta\right).$$

## B. Proofs

### B.1. Proofs of Section 3

#### B.1.1. PROOF OF THEOREM 3.1

We first analyze whether the true model $m^*$ remains in the hypothesis class for all $T$ rounds. To see this, by Lemma A.2, for all $i \in S_t$ and $t \in [T]$, we have

$$\sum_{o\in\mathcal{D}_t} \log(\mathbb{P}_i^\pi(o)) - \beta \leq \sum_{o\in\mathcal{D}_t} \log(\mathbb{P}_{m^*}^\pi(o)),$$

where $\beta = \log(MT/\delta)$. Hence, due to our construction of the next hypothesis set in Algorithm 2, with probability $1 - \delta/T$, $m^* \in S_{t+1}$. As the worst-case classification round does not exceed $M$ with Assumption 1, without loss of generality, we assume that $T = O(M)$.

Next, for every $t^{th}$ round, we prove that $S_{t+1} \subseteq S_t/\bar{S}_{t,\lambda}^{\pi_t}(m^*)$ where

$$\bar{S}_{t,\lambda}^\pi(m^*) = \{i \in S_t | D_{\mathrm{H}}(\mathbb{P}_i^{\pi_t}, \mathbb{P}_{m^*}^{\pi_t}) \geq \lambda\}.$$

Note that with probability $1 - \delta/T$,

$$0 \geq \sum_{o\in\mathcal{D}_t} \log(\mathbb{P}_{m^*}^\pi(o)) - \sum_{o\in\mathcal{D}_t} \log(\mathbb{P}_{\hat{m}_t}^\pi(o)) \geq -\beta,$$

for all $i \in S_t$. From Lemma A.3, for all $i \in S_{t+1}$, by taking union bound, it must satisfy that

$$\beta \geq \sum_{o\in\mathcal{D}_t} \log\left(\frac{\mathbb{P}_{m^*}^\pi(o)}{\mathbb{P}_i^\pi(o)}\right) \geq 2N_{\mathrm{cls}} \cdot D_{\mathrm{H}}^2(\mathbb{P}_i^{\pi_t}, \mathbb{P}_{m^*}^{\pi_t}) - 3\beta,$$

where the first inequality holds due to our construction of $S_{t+1}$. Thus, for all $i \in S_{t+1}$, we must have

$$D_{\mathrm{H}}^2(\mathbb{P}_i^{\pi_t}, \mathbb{P}_{m^*}^{\pi_t}) \leq \frac{2\beta}{N_{\mathrm{cls}}}, \quad \forall i \in S_{t+1}.$$

This means with $N_{\text{cls}} > 2\frac{\log(M/\delta)}{\lambda^2}$ test samples per round, $S_{t+1} \subseteq S_t/\bar{S}_{t,\lambda}^{\pi_t}(m^*)$.

Finally, with our design of $\pi_t$, we always choose $\pi_t$ such that

$$|\bar{S}_{t,\lambda}^{\pi_t}(m^*)| \geq C_\lambda(\Pi_{\mathcal{C}})^{-1} \cdot |S_t|.$$

This implies with probability at least $1 - \delta/T$, we always have

$$\frac{|S_{t+1}|}{|S_t|} \leq 1 - C_\lambda(\Pi_{\mathcal{C}})^{-1},$$

which translates to

$$\mathbb{E}\left[\frac{|S_{t+1}|}{|S_t|}\Big|S_t\right] \leq 1 - \frac{1}{2}C_\lambda(\Pi_{\mathcal{C}})^{-1}.$$

Note that in the worst case, the ratio remains 1 with probability less than $\delta/T$. Let $W_t := \left(1 + \frac{1}{2}C_\lambda(\Pi_{\mathcal{C}})^{-1}\right)^t |S_t|$. Then $\{W_t\}_{t\geq 0}$ is a super-martingale, and thus, by Lemma A.1, we have

$$\left(1 + \frac{1}{2}C_\lambda(\Pi_{\mathcal{C}})^{-1}\right)^T |S_T| \leq \frac{1}{\delta}|S_0|,$$

with probability at least $1 - \delta$. Under this success event, as soon as $T > 2C_\lambda(\Pi_{\mathcal{C}}) \cdot \log(M/\delta)$, we must have $|S_T| = 1$.

To summarize, if we use $N_{\text{cls}} = O(\lambda^{-2} \cdot \log(M/\delta))$ samples per classification round for $T = O(C_\lambda(\Pi_{\mathcal{C}}) \cdot \log(M/\delta))$ rounds, the algorithm terminates with the correct task identifier $m^*$ with probability at least $1 - \delta$, concluding the proof.

### B.1.2. PROOF OF LEMMA 3.2

Following the definition of DEC in (6), we have that

$$\text{dec}_\gamma(\mathbb{M}) \geq \max_{S\in 2^{[M]}} \min_{\pi\sim\Delta([K])} \max_{i\in S} \mathbb{E}_{k\sim\pi}[\Delta_i(k)] - \gamma\mathbb{E}_{k\sim\pi,m\sim\mathcal{U}(S)}[D_{\text{H}}^2(\nu_i(k),\nu_m(k))].$$

Recall that the randomized coefficient in (3) can be rewritten as the following:

$$\widetilde{C}(\mathbb{M}) = \left(\min_{S\in 2^{[M]},|S|>1} \max_{\pi\sim\Delta(\mathcal{A}_\lambda)} \min_{i\in S} \mathbb{E}_{m\sim\mathcal{U}(S)}[\mathbb{E}_{k\sim\pi}\left[\mathbf{1}\{\mu_i(k) \neq \mu_m(k)\}\right]]\right)^{-1},$$

and let $S_{adv}$ be the outer solution of the above min-max-min optimization. Now for any $\pi \in \Delta([K])$, let $i^*(\pi)$ be the one that achieves

$$i^*(\pi) := \arg\min_{i\in S_{adv}} \mathbb{E}_{m\sim\mathcal{U}(S)}[\mathbb{E}_{k\sim\pi}\left[\mathbf{1}\{\mu_i(k) \neq \mu_m(k)\}\right]]. \tag{8}$$

We claim that there must exist $\bar{i}(\pi) \in S_{adv}/\{i^*(\pi)\}$ such that the following holds:

$$\mathbb{E}_{m\sim\mathcal{U}(S)}[\mathbb{E}_{k\sim\pi}\left[\mathbf{1}\{\mu_{\bar{i}}(k) \neq \mu_m(k)\}\right]] \leq 4\mathbb{E}_{m\sim\mathcal{U}(S)}\left[\mathbb{E}_{k\sim\pi}[\mathbf{1}\{\mu_{i^*(\pi)}(k) \neq \mu_m(k)\}]\right]. \tag{9}$$

To see this, note that

$$\mathbf{1}\{\mu_{\bar{i}}(k) \neq \mu_m(k)\} \leq \mathbf{1}\{\mu_{\bar{i}}(k) \neq \mu_{i^*(\pi)}(k)\} + \mathbf{1}\{\mu_{i^*(\pi)}(k) \neq \mu_m(k)\},$$

and then, by taking $\bar{i}(\pi) := \arg\min_{i\in S_{adv}/\{i^*(\pi)\}} \mathbb{E}_{k\sim\pi}[\mathbf{1}\{\mu_{i^*(\pi)}(k) \neq \mu_{\bar{i}}(k)\}]$, we can verify that

$$\mathbb{E}_{k\sim\pi}[\mathbf{1}\{\mu_{i^*(\pi)}(k) \neq \mu_{\bar{i}}(k)\}] \leq 2\mathbb{E}_{m\sim\mathcal{U}(S_{adv})}\left[\mathbb{E}_{k\sim\pi}[\mathbf{1}\{\mu_{i^*(\pi)}(k) \neq \mu_m(k)\}]\right],$$

since $|S_{adv}| > 1$ and the indicator function is nonnegative. Note that for all $\pi \in \Delta(\mathcal{A}_\lambda)$, by construction, $\mathbb{E}_{m\sim\mathcal{U}(S)}\left[\mathbb{E}_{a\sim\pi}[\mathbf{1}\{\mu_{i^*(\pi)}(a) \neq \mu_m(a)\}]\right] \leq \widetilde{C}(\mathbb{M})^{-1}$.

Now going back to the DEC lower-bound, we have

$$\mathtt{dec}_\gamma(\mathbb{M}) \geq \min_{\pi \in \Delta([K])} \max_{i \in S_{adv}} \mathbb{E}_{k \sim \pi}[\Delta_i(k)] - \gamma \mathbb{E}_{k \sim \pi}[\mathbb{E}_{m \sim \mathcal{U}(S_{adv})}[D_{\mathrm{H}}^2(\nu_i(k), \nu_m(k))]]$$

$$\geq \min_{\pi \in \Delta([K])} \max_{i \in S_{adv}} \underbrace{\sum_{k \in \{k_m^*\}_m} \Delta_i(k) \cdot \pi(k) + \frac{1}{8}\pi(k \in \mathcal{A}_\lambda)}_{I} \tag{10}$$

$$- \gamma \left( \underbrace{200\epsilon^2 \cdot \pi(k \notin \mathcal{A}_\lambda) + \mathbb{E}_{m \sim \mathcal{U}(S_{adv})} \left[ \mathbb{E}_{k \sim \pi_\lambda}[D_{\mathrm{H}}^2(\nu_i(k), \nu_m(k))] \right] \cdot \pi(k \in \mathcal{A}_\lambda)}_{II} \right), \tag{11}$$

where we define $\pi_\lambda = \pi(\cdot | k \in \mathcal{A}_\lambda)$. Now for every $\pi$ and the corresponding $\pi_\lambda$, let $i^*(\pi_\lambda)$ as defined in (8) and $\bar{i}(\pi_\lambda) = S_{adv}/\{i^*(\pi_\lambda)\}$. Now we either choose $i = i^*(\pi_\lambda)$ if

$$\pi(k_{i^*(\pi_\lambda)}^*) < \pi(k_{\bar{i}(\pi_\lambda)}^*),$$

and $\bar{i}(\pi_\lambda)$ in the other case. We divide into two cases.

**1.** $\pi(k_{i^*(\pi_\lambda)}^*) < \pi(k_{\bar{i}(\pi_\lambda)}^*)$**:**   In the former case, note that for all $m \neq i^*(\pi_\lambda)$,

$$\Delta_{i^*(\pi_\lambda)}(k_m^*) \geq 10\epsilon,$$

and therefore,

$$\sum_{a \in \{k_m^*\}_m} \Delta_{i^*(\pi_\lambda)}(k)\pi(k) \geq 5\epsilon \pi(k \notin \mathcal{A}_\lambda).$$

Therefore, we have $I \geq 5\epsilon \pi(k \notin \mathcal{A}_\lambda) + \frac{1}{8}\pi(k \in \mathcal{A}_\lambda)$ in (11).

For the second term, note that

$$\mathbb{E}_{m \sim \mathcal{U}(S_{adv})} \left[ \mathbb{E}_{k \sim \pi_\lambda}[D_{\mathrm{H}}^2(\nu_{i^*(\pi_\lambda)}(k), \nu_m(k))] \right] \leq \lambda^2 \mathbb{E}_{m \sim \mathcal{U}(S_{adv})} \left[ \mathbb{E}_{k \sim \pi_\lambda}[\mathbf{1}\{\nu_{i^*(\pi_\lambda)}(k), \nu_m(k))\}] \right] \leq \lambda^2 \widetilde{C}(\mathbb{M})^{-1}.$$

Therefore, the second term becomes $II \leq 200\epsilon^2 \pi(k \notin \mathcal{A}_\lambda) + \lambda^2 \widetilde{C}(\mathbb{M})^{-1}\pi(k \in \mathcal{A}_\lambda)$.

**2.** $\pi(k_{i^*(\pi_\lambda)}^*) > \pi(k_{\bar{i}(\pi_\lambda)}^*)$**:**   In the latter case, repeat the same process except that now we take the worst-case inner-instance $i = \bar{i}(\pi_\lambda)$, we get the same inequalities.

Combining all results, we can conclude that

$$I - \gamma II \geq (5\epsilon - 200\epsilon^2 \gamma)\pi(k \notin \mathcal{A}_\lambda) + \left( \frac{1}{8} - \gamma \lambda^2 \tilde{C}(\mathbb{M})^{-1} \right) \pi(k \in \mathcal{A}_\lambda) > 3\epsilon,$$

for any $\pi \in \Delta([K])$ with $\gamma \leq c_\gamma \min \left( \epsilon^{-1}, \lambda^{-2}\widetilde{C}(\mathbb{M}) \right)$ for some sufficiently small $c_\gamma > 0$. Therefore,

$$\mathtt{dec}_\gamma(\mathbb{M}) > 3\epsilon,$$

concluding the proof.

### B.1.3. PROOF OF THEOREM 3.3

To identify the optimal arm (so that we can play it for the majority of rounds), it must hold $\mathtt{dec}_\gamma(\mathcal{M}) < \epsilon$. On the other hand, we have the following lower bound, which is a reminiscent of lower bound results in (Chen et al., 2022a) and (Foster et al., 2021):

**Theorem B.1.** *For any $\delta \in (0, 1)$ and a regret minimization algorithms for $H$ rounds,*

$$\mathrm{Reg}_H(\mathbb{M}) \geq C_2 \cdot \max_{\gamma \geq C_1 \cdot \sqrt{H}} \min \left( (\mathtt{dec}_\gamma(\mathbb{M}) - \delta) \cdot H, \gamma \right),$$

*with probability at least $\delta$ for some absolute constant $C_1, C_2 > 0$.*

Thus, we must have $\gamma = \tilde{\Omega}(\lambda^{-2}\tilde{C}(\mathbb{M}))$ so that we can have $\text{dec}_\gamma(\mathbb{M}) < 3\epsilon$ for all $\gamma$ greater than this threshold. Otherwise, any algorithm must suffer from at least $\tilde{\Omega}(\min(\epsilon H, \lambda^{-2}\tilde{C}(\mathbb{M})))$ regret with probability at least $\delta = 1/H \ll \epsilon$. Furthermore, since $\text{Reg}_H \geq \text{Reg}_{H_0}$ for any $H \geq H_0$, it holds that for all $H \geq H_0 = \lambda^{-4}\tilde{C}(\mathbb{M})^2$, we must suffer $\text{Reg}_H = \tilde{\Omega}(\lambda^{-2}\tilde{C}(\mathbb{M}))$.

### B.1.4. PROOF OF THEOREM B.1

The proof follows Section C.1 in (Foster et al., 2021) with minor modification. Let us define a regret for individual instance:

$$\text{Reg}_H^m := \sum_{t=1}^{H} \mu_m(k_m^*) - \mu_m(k_t).$$

Let $\mathcal{E}_m$ an event such that $\{\text{Reg}_H^m \leq c_1\gamma\}$ with some sufficiently small constant $c_1$. For any algorithm, $\gamma > 0$ and $\delta = 1/H$ we consider, we assume that for all $m \in [M]$, $\mathbb{P}_m(\mathcal{E}_m) \geq 1 - \delta$ since otherwise the algorithm suffers from at least $\gamma$ regret with probability at least $\delta$.

Let us fix an algorithm $\mathcal{A}$ such that at $t^{th}$ round with previous observations $\mathcal{H}^{t-1} = (o_1, ..., o_{t-1})$ where $o_t = (x_t, a_t, r_t)$, and the policy at each round is decided by an algorithm $\pi_t = \mathcal{A}(\cdot|x_t, \mathcal{H}^{(t-1)})$. Let $\mathbb{P}_m^H$ be the distribution of sequential observations $(o_1, ..., o_H)$ for $H$ rounds with bandit $\nu_m$. Following Lemmas are adapted from (Foster et al., 2021):

**Lemma B.2** (Lemma A.11 in Foster et al. 2021). *For any two distributions $\mu, \nu$ on a measurable space $\mathcal{X}$, and any bounded real-valued function $h : \mathcal{X} \to \mathbb{R}$ with $0 \leq h(X) \leq B$, we have*

$$|\mathbb{E}_\mu[h(X)] - \mathbb{E}_\nu[h(X)]| \leq \sqrt{2B(\mathbb{E}_\mu[h(X)] + \mathbb{E}_\nu[h(X)]) \cdot D_{\text{H}}^2(\mu, \nu)}.$$

*In particular,*

$$|\mathbb{E}_\mu[h(X)] - \mathbb{E}_\nu[h(X)]| \leq 3\mathbb{E}_\nu[h(X)] + 4BD_{\text{H}}^2(\mu, \nu).$$

**Lemma B.3** (Lemma A.13 in Foster et al. 2021). *For any two bandit instances $\nu_i, \nu_j \in \mathbb{M}$,*

$$D_{\text{H}}^2(\mathbb{P}_i^H, \mathbb{P}_j^H) \leq C_H \sum_{t=1}^{H} \mathbb{E}_i[\mathbb{E}_{k\sim\pi_t}[D_{\text{H}}^2(\nu_i(k), \nu_j(k))]],$$

*where $C_H > 0$ is a sufficiently large absolute constant.*

Given the lemmas, for any $\omega \in \Delta([M])$ and for any algorithm that generates an adaptive policy $\pi_t$, let $\hat{\pi} := \frac{1}{H}\sum_{t=1}^{H} \pi(\cdot|\mathcal{H}^{(t-1)})$ (note that this is a random variable), and let $\bar{\pi} := \mathbb{E}_{m\sim\omega}[\hat{\pi}]$.

**Lemma B.4** (Minor Edit of Lemma C.1 in Foster et al. 2021). *For any two bandit instances $\nu_i, \nu_j \in \mathbb{M}$,*

$$\frac{1}{H}\mathbb{E}_j[\text{Reg}_H^i \cdot \mathbf{1}\{\mathcal{E}_i^c\}] \lesssim \frac{c_1\gamma}{H} \cdot D_{\text{H}}^2(\mathbb{P}_i^H, \mathbb{P}_j^H) + \sqrt{D_{\text{H}}^2(\mathbb{P}_i^H, \mathbb{P}_j^H)\mathbb{E}_i[\mathbb{E}_{k\sim\hat{\pi}}[D_{\text{H}}^2(\nu_i(k), \nu_j(k))]]} + \delta.$$

We start with the following inequality for a prior $\omega$ such that:

$$\sup_{m\in[M]} \mathbb{E}_{a\sim\bar{\pi}}[\nu_m(a_m^*) - \nu_m(a)] - \gamma \cdot \mathbb{E}_{\bar{m}\sim\omega}[\mathbb{E}_{a\sim\bar{\pi}}[D_H^2(\nu_{\bar{m}}(a), \nu_m(a))]] \geq \text{dec}_\gamma(\mathbb{M}).$$

Such a prior $\omega \in \Delta([M])$ must exist due to the definition of $\text{dec}_\gamma$. Note that

$$H \cdot \mathbb{E}_{a\sim\bar{\pi}}[\nu_m(a_m^*) - \nu_m(a)] = \mathbb{E}_{\bar{m}\sim\omega}\mathbb{E}_{a\sim\hat{\pi}}[\nu_m(a_m^*) - \nu_m(a)] = H \cdot \mathbb{E}_{\bar{m}\sim\omega}[\text{Reg}_H^m]$$

$$= \sum_{\bar{m}} \omega_{\bar{m}}\mathbb{E}_{\bar{m}}[\text{Reg}_H^m] = \sum_{\bar{m}} \omega_{\bar{m}} \left( \underbrace{\mathbb{E}_{\bar{m}}[\text{Reg}_H^m \cdot \mathbf{1}\{\mathcal{E}_m\}]}_{I} + \underbrace{\mathbb{E}_{\bar{m}}[\text{Reg}_H^m \cdot \mathbf{1}\{\mathcal{E}_m^c\}]}_{II} \right).$$

For $I$, we apply Lemma B.2 to get

$$I \leq 3\mathbb{E}_m[\text{Reg}_H^m \cdot \mathbf{1}\{\mathcal{E}_m\}] + 4\gamma D_{\text{H}}^2(\mathbb{P}_{\bar{m}}^H, \mathbb{P}_m^H) \leq 3\mathbb{E}_m[\text{Reg}_H^m] + 4\gamma D_{\text{H}}^2(\mathbb{P}_{\bar{m}}^H, \mathbb{P}_m^H).$$

For $II$, we apply Lemma B.4 to get

$$II \lesssim (H\epsilon + c_1\gamma)D_{\mathrm{H}}^2(\mathbb{P}_{\bar{m}}^H, \mathbb{P}_m^H) + H\sqrt{D_{\mathrm{H}}^2(\mathbb{P}_{\bar{m}}^H, \mathbb{P}_m^H) \cdot \mathbb{E}_{\bar{m}}[\mathbb{E}_{k\sim\hat{\pi}}[D_{\mathrm{H}}^2(\nu_{\bar{m}}(k), \nu_m(k))]]} + H\delta.$$

Combining these inequalities, we have

$$\mathbb{E}_m[\mathrm{Reg}_H^m] \gtrsim H \cdot \mathrm{dec}_\gamma(\mathbb{M}) - \sum_{\bar{m}} \omega_{\bar{m}} \cdot \left(c_1\gamma D_{\mathrm{H}}^2(\mathbb{P}_{\bar{m}}^H, \mathbb{P}_m^H) + H\sqrt{D_{\mathrm{H}}^2(\mathbb{P}_{\bar{m}}^H, \mathbb{P}_m^H) \cdot \mathbb{E}_{\bar{m}}[\mathbb{E}_{a\sim\hat{\pi}}[D_{\mathrm{H}}^2(\nu_{\bar{m}}(a), \nu_m(a))]]}\right)$$
$$+ \gamma H \cdot \mathbb{E}_{\bar{m}\sim\omega}[\mathbb{E}_{a\sim\bar{\pi}}[D_{\mathrm{H}}^2(\nu_{\bar{m}}(a), \nu_m(a))]] - H\delta.$$

On the other hand, we can apply Lemma B.3 to bound that

$$D_{\mathrm{H}}^2(\mathbb{P}_{\bar{m}}^H, \mathbb{P}_m^H) \leq C_H \sum_{t=1}^H \mathbb{E}_{\bar{m}}[\mathbb{E}_{k\sim\pi_t}[D_{\mathrm{H}}^2(\nu_{\bar{m}}(k), \nu_m(k))]]$$
$$= C_H H \cdot \mathbb{E}_{\bar{m}}[\mathbb{E}_{a\sim\hat{\pi}}[D_{\mathrm{H}}^2(\nu_{\bar{m}}(a), \nu_m(a))]] = C_H H \cdot \mathbb{E}_{a\sim\bar{\pi}}[D_{\mathrm{H}}^2(\nu_{\bar{m}}(a), \nu_m(a))].$$

Plugging these results, we have

$$\mathbb{E}_m[\mathrm{Reg}_H^m] \gtrsim H \cdot \mathrm{dec}_\gamma(\mathbb{M}) - H(c_1\gamma + \sqrt{H}) \cdot \sum_{\bar{m}} \omega_{\bar{m}}\mathbb{E}_{\bar{\pi}}[D_{\mathrm{H}}^2(\nu_{\bar{m}}(k), \nu_m(k))]$$
$$+ \gamma H \cdot \mathbb{E}_{\bar{m}\sim\omega}[\mathbb{E}_{k\sim\bar{\pi}}[D_{\mathrm{H}}^2(\nu_{\bar{m}}(k), \nu_m(k))]] - H\delta.$$

Note that

$$\mathbb{E}_{\bar{m}\sim\omega}[\mathbb{E}_{a\sim\bar{\pi}}[D_{\mathrm{H}}^2(\nu_{\bar{m}}(k), \nu_m(k))]] = \sum_{\bar{m}} \omega_{\bar{m}}\mathbb{E}_{\bar{\pi}}[D_{\mathrm{H}}^2(\nu_{\bar{m}}(k), \nu_m(k))].$$

This implies that as long as $c_1$ is a sufficiently small constant and $\gamma \gtrsim \sqrt{H}$, the expected lower bound is given by

$$\mathbb{E}_m[\mathrm{Reg}_H^m] \gtrsim H\left(\mathrm{dec}_\gamma(\mathbb{M}) - \delta\right).$$

*Proof of Lemma B.3.* The general version of subadditivity lemma in (Foster et al., 2021) is stated as the following:

**Lemma B.5.** *Let $(\mathcal{X}_1, \mathcal{F}_1), ..., (\mathcal{X}_n, \mathcal{F}_n)$ be a sequence of measurable spaces, and let $\mathcal{X}^{(i)} = \Pi_{t=1}^i \mathcal{X}_t$ and $\mathcal{F}^{(i)} = \bigotimes_{t=1}^i \mathcal{F}_t$. For each $i$, let $\mu^{(i)}, \nu^{(i)}$ be probability kernels from $(\mathcal{X}^{(i-1)}, \mathcal{F}^{(i-1)})$ to $(\mathcal{X}^{(i)}, \mathcal{F}^{(i)})$. Let $\mu, \nu$ be the laws of sequence $X_1, ..., X_n$ following the sequence of $(\mu^{(1)}, ..., \mu^{(n)}), (\nu^{(1)}, ..., \nu^{(n)})$ respectively. Then it holds that*

$$D_{\mathrm{H}}(\mu, \nu) \leq 10^2 \log(n) \cdot \mathbb{E}_\mu[\sum_{i=1}^n D_{\mathrm{H}}^2(\mu^{(i)}(\cdot|X_1, ..., X_{i-1}), \nu^{(i)}(\cdot|X_1, ..., X_{i-1}))].$$

*Furthermore, if there exists a constant $V$ such that $\sup_{(x_1,...,x_{i-1})\in\mathcal{X}^{(i-1)}} \sup_{o_i\in\mathcal{F}_i} \frac{\mu^{(i)}(o_i|x_1,...,x_{i-1})}{\nu^{(i)}(o_i|x_1,...,x_{i-1})}$ for all $i$, then*

$$D_{\mathrm{H}}(\mu, \nu) \leq 3\log(V) \cdot \mathbb{E}_\mu[\sum_{i=1}^n D_{\mathrm{H}}^2(\mu^{(i)}(\cdot|X_1, ..., X_{i-1}), \nu^{(i)}(\cdot|X_1, ..., X_{i-1}))].$$

Our construction belongs to the latter case, since the probability of observing $r_t = 1$ or $r_t = 0$ is larger than $\frac{1-\lambda}{2} \geq 1/4$ for any $\lambda \leq 1/2$. $\square$

*Proof of Lemma B.4.* In our construction, for all pair of bandit instances $\mu, \nu \in \mathbb{M}$, the optimal values are the same, that is,

$$\mu(k_\mu^*) - \nu(k_\nu^*) = 0,$$

where $k_\mu^*, k_\nu^*$ are the optimal actions for $\mu, \nu$ respectively. The remaining steps are identical to the proof in (Foster et al., 2021) (see their Section C.1.2), and we omit them here. $\square$

## B.2. Proofs of Section 4

### B.2.1. PROOF OF LEMMA 4.1

*Proof.* We can rework the result (Hsu et al., 2011, Theorem 1), originally designed for the excess quadratic loss, to write

$$\mathbb{P}\left(\mathbb{E}_{\mathcal{P}}\left[|x^\top \hat{\theta}_{ik} - x^\top \theta_{ik}|\right] > \sqrt{\frac{5\sigma^2(d + 2\sqrt{d\log(2/\delta)} + 2\log(2/\delta))}{N}}\right) \le \delta$$

where $\hat{\theta}$ is the ordinary least squares with $N$ samples. Then, we just plug $\delta = \frac{1}{2HMK}$ in the expression to obtain the guarantee with a few algebraic manipulations. $\qquad\square$

### B.2.2. PROOF OF THEOREM 4.2

Let us start looking at the sample complexity. Since the Algorithm 3 takes $N_{\text{est}}$ samples for every arm $k \in [K]$ and simulator $\nu_i \in \mathbb{M}$, we can conclude that the statistical complexity of meta training is $\frac{4MK\log(4HMK)}{\min(\Delta_{\min}^2, \lambda^2)}$.

Assuming access to parallel simulators, the computational cost of meta training depends on the cost of executing line 12 in Algorithm 3, which is calling Algorithm 4. The latter requires executing $|S|$ evaluations at lines 5, 6, where $|S| \le M$, and to compute the greedy step (line 3), a cost that is paid for every call to the recursive procedure (line 8). Computing the greedy step through Algorithm 5 is done in $4K/\lambda^4$ steps. Finally, we can bound the number of calls to the recursive procedure with the total number of nodes in the tree, which is $\mathcal{O}(M^2)$. Putting all together we get a complexity of order $\mathcal{O}(M^3K/\lambda^4)$.

### B.2.3. PROOF OF LEMMA 4.3

*Proof.* The result follows directly from the approximation guarantee of the greedy algorithm to build the decision tree (Arkin et al., 1993), which guarantees $d = \mathcal{O}(\log M + 1)C_\lambda^*(\mathbb{M})$. Especially, we have to prove that the previous guarantee does not degrade with our implementation, which include a $\lambda/4$-discretization of the space of tests (see Algorithm 5, line 4). Thanks to the separation condition (Assumption 2), we can prove that every test $\hat{\mu}(k) \le b$ with $b \in [0, 1]$ can be replicated with *at most* two tests defined on the discretized space, i.e., $\hat{\mu}(k) \le b$ with $b \in [0, 1]_{\lambda/4}$. Since the approximation degrades of a constant factor only, the result $D = \mathcal{O}(\log M + 1)C_\lambda^*(\mathbb{M})$ holds. $\qquad\square$

### B.2.4. PROOF OF THEOREM 4.4

*Proof.* To derive the upper bound on the regret, we aim to prove that the remaining task $\hat{\nu}_{m^*}$ at the end of the *Explicit Classify* phase corresponds, up to a small estimation error, to the true test task $\nu^*$ with high probability, and that the policy $\pi^*$ played from there on in the *Exploit* phase corresponds to the optimal policy for the test task $\nu^*$ with high probability (despite the mentioned estimation error).

If we let $\pi^*(x) = \arg\max_{\pi \in \Pi} x^\top \theta^*_{\pi(x)}$ the optimal policy of the (true) test task, we aim to prove

$$\mathbb{P}_{\mathcal{P}}(\hat{\pi}^*(x) \ne \pi^*(x)) = \mathbb{P}(\text{``\textit{Explicit Classify} fails''} \lor \text{``\textit{Exploit} fails''}) \le 1/H$$

which we can guarantee by showing that the *Explicit Classify* and *Exploit* phases fail with probability less than $1/2H$ and then applying a union bound.

Let us first take the good event for the *Explicit Classify* phase, which means the remaining $\hat{\nu}_{m^*}$ is a "good" estimate of the test task $\nu^*$. We have that

$$\mathbb{P}(\text{``\textit{Exploit} fails''}) = \mathbb{P}_{\mathcal{P}}\left(\hat{\pi}^*(x) \ne \pi^*(x)\right) \tag{12}$$

$$\le \mathbb{P}_{\mathcal{P}}\left(\bigcup_{i \in [M]} \bigcup_{k \in [K]} x^\top \hat{\theta}_{i\pi^*(x)} \le x^\top \hat{\theta}_{ik}\right) \tag{13}$$

$$\le \sum_{i \in [M]} \sum_{k \in [K]} \mathbb{P}_{\mathcal{P}}\left(x^\top \hat{\theta}_{i\pi^*(x)} \le x^\top \hat{\theta}_{ik}\right) \le \sum_{i \in [M]} \sum_{k \in [K]} \frac{1}{2HMK} \le \frac{1}{2H} \tag{14}$$

where we consider any possible choice of the remaining task $\hat{\nu}_{m^*}$ and the test task $\nu^*$ to write (13) from (12), we apply a union bound and the estimation guarantee of Algorithm 3 (see Lemma 4.1) to write (14).

Conversely, under the good event for the *Exploit* phase we aim to prove that the *Explicit Classify* phase fails with probability less than $1/2H$. Since the *Explicit Classify* phase is actually a sequence of tests, we need to bound the probability that each test fails. Formally, let $J$ denote the number of iterations of the loop between lines 3-11 (Algorithm 6), through a union bound we have

$$\mathbb{P}(\text{``Explicit Classify fails''}) = \mathbb{P}\left(\bigcup_{j\in[J]} \text{``test at iteration } j \text{ fails''}\right) \leq \sum_{j\in[J]} \mathbb{P}(\text{``test at iteration } j \text{ fails''})$$

Now, we need to design $N_{\text{cls}}$ such that the test at each iteration fails with probability less than $\frac{1}{2HJ} \geq \frac{1}{2HD}$ where $D$ is the depth of $\text{tree}(\hat{\mathbb{M}})$. For each iteration $j$, take the test $\mu_k \leq b$ and let $\overline{\mu} = \frac{1}{N_{\text{cls}}}\sum_{n\in[N_{\text{cls}}]} r_n$ the empirical mean of the samples $r_n \sim \nu^*(x_n, k)$ collected from the test task at line 5 (Algorithm 6). We need to assure that the event of $\overline{\mu}$ falling on one side of the test while the "right" $\tilde{\mu}_k$ is on the other side (see lines 6-11 of Algorithm 6) happens with small enough probability. Formally,

$$\begin{aligned}\mathbb{P}(\text{``test at iteration } j \text{ fails''}) &= \mathbb{P}(\{\overline{\mu} \leq b \wedge \tilde{\mu}_k > b + \lambda\} \cup \{\overline{\mu} > b \wedge \tilde{\mu}_k \leq b - \lambda\}) \\ &\leq \mathbb{P}(|\overline{\mu} - \tilde{\mu}_k| > \lambda) \\ &\leq \mathbb{P}(|\overline{\mu} - \mu_k| > \lambda/2) + \mathbb{P}(|\tilde{\mu}_k - \mu_k| > \lambda/2)\end{aligned}$$

For the second event, we invoke the estimation guarantee of Algorithm 3 (see Lemma 4.1) to write $\mathbb{P}(|\tilde{\mu}_k - \mu_k| > \lambda/2) \leq \frac{1}{2HMK} \leq \frac{1}{4HD}$. For the first event, we need to assure that $\mathbb{P}(|\overline{\mu} - \mu_k| > \lambda/2) \leq \frac{1}{4HD}$. Since $\overline{\mu}$ is the empirical mean of $\mu_k$, by applying the Hoeffding's inequality, we have that $N_{\text{cls}} \geq \frac{2\log(8HD)}{\lambda^2}$ gives the desired guarantee.

Having demonstrated that $\mathbb{P}_{\mathcal{P}}(\hat{\pi}^*(x) \neq \pi^*(x))$ holds with probability less than $1/H$, we can finally write

$$\text{Reg}_H(\mathbb{M}) = \mathbb{E}_{\mathcal{P}}\left[\sum_{t=1}^{JN_{\text{cls}}} \max_{k\in[K]} x_t^\top \theta_k^* - r_t\right] + \mathbb{E}_{\mathcal{P}}\left[\sum_{t=JN_{\text{cls}}+1}^{H} \max_{k\in[K]} x_t^\top \theta_k^* - x_t^\top \theta_{\hat{\pi}^*(x_t)}^*\right] \leq \frac{2D\log(8HD)}{\lambda^2}$$

by taking $x_t^\top \theta_k^* - x_t^\top \theta_{\hat{\pi}^*(x_t)}^* = 0$ in the good event, upper bounding $\max_{k\in[K]} x_t^\top \theta_k^* - r_t \leq 1$ and $JN \leq DN_{\text{cls}}$, and then apply the approximation guarantee $D = \mathcal{O}((\log M + 1)C_\lambda^*(\mathbb{M}))$ from Lemma 4.3 to get the result. $\qquad\square$

## B.3. Proof of Auxiliary Lemmas

### B.3.1. PROOF OF LEMMA A.2

The proof of MLE-based confidence set construction is by now standard and can be found in several prior works (*e.g.*, Liu et al. 2022). We adapt the proofs from (Kwon et al., 2024) for completeness.

*Proof.* The proof follows a Chernoff bound type of technique:

$$\begin{aligned}\mathbb{P}_{\nu^*}&\left(\sum_{o\in\mathcal{D}} \log\left(\frac{\mathbb{P}_\nu^\pi(o)}{\mathbb{P}_{\nu^*}^\pi(o)}\right) \geq \mathbb{E}_{\nu^*}\left[\sum_{o\in\mathcal{D}} \log\left(\frac{\mathbb{P}_\nu^\pi(o)}{\mathbb{P}_{\nu^*}^\pi(o)}\right)\right] + \beta\right) \\ &\leq \mathbb{P}_{\nu^*}\left(\exp\left(\sum_{o\in\mathcal{D}} \log\left(\frac{\mathbb{P}_\nu^\pi(o)}{\mathbb{P}_{\nu^*}^\pi(o)}\right)\right) \geq \exp(\beta)\right) \\ &\leq \mathbb{E}_{\nu^*}\left[\exp\left(\sum_{o\in\mathcal{D}} \log\left(\frac{\mathbb{P}_\nu^\pi(o)}{\mathbb{P}_{\nu^*}^\pi(o)}\right)\right)\right] \exp(-\beta).\end{aligned}$$

The last inequality is by the Markov's inequality. Note that random variables are $o$ in the trajectory dataset $\mathcal{D}$, and

$$\mathbb{E}_{\nu^*}\left[\sum_{o\in\mathcal{D}} \log\left(\frac{\mathbb{P}_\nu^\pi(o)}{\mathbb{P}_{\nu^*}^\pi(o)}\right)\right] = -\text{KL}(\mathbb{P}_{\nu^*}(\mathcal{D})||\mathbb{P}_\nu(\mathcal{D})) \leq 0.$$

Furthermore,

$$\mathbb{E}_{\nu^*}\left[\exp\left(\sum_{o\in\mathcal{D}}\log\left(\frac{\mathbb{P}^\pi_\nu(o)}{\mathbb{P}^\pi_{\nu^*}(o)}\right)\right)\right] = \mathbb{E}_{\nu^*}\left[\Pi_{o\in\mathcal{D}}\frac{\mathbb{P}^\pi_\nu(o)}{\mathbb{P}^\pi_{\nu^*}(o)}\right] = 1.$$

Combining the above, taking a union bound over $\nu\in\mathbb{M}$, letting $\beta=\log(M/\delta)$, with probability $1-\delta$, the inequality in Lemma A.2 holds. $\qquad\square$

### B.3.2. PROOF OF LEMMA A.3

*Proof.* By the TV-distance and Hellinger distance relation, for any $\iota, \tau, \pi$ and $t\in[H]$,

$$D^2_{\mathrm{H}}\left(\mathbb{P}^\pi_\nu, \mathbb{P}^\pi_{\nu^*}\right) = 1 - \mathbb{E}_{o\sim\mathbb{P}^\pi_{\nu^*}}\left[\sqrt{\frac{\mathbb{P}^\pi_\theta(o)}{\mathbb{P}^\pi_{\theta^*}(o)}}\right] \leq -\log\left(\mathbb{E}_{o\sim\mathbb{P}^\pi_{\nu^*}}\left[\sqrt{\frac{\mathbb{P}^\pi_\nu(o)}{\mathbb{P}^\pi_{\nu^*}(o)}}\right]\right).$$

By the Chernoff bound,

$$\mathbb{P}_{\nu^*}\left(\sum_{o\in\mathcal{D}}\log\left(\sqrt{\frac{\mathbb{P}^\pi_\nu(o)}{\mathbb{P}^\pi_{\nu^*}(o)}}\right) \geq |\mathcal{D}|\cdot\log\mathbb{E}_{o\sim\mathbb{P}^\pi_{\nu^*}}\left[\sqrt{\frac{\mathbb{P}^\pi_\nu(o)}{\mathbb{P}^\pi_{\nu^*}(o)}}\right] + \beta\right)$$

$$\leq \mathbb{E}_{\nu^*}\left[\frac{\exp\left(\sum_{o\in\mathcal{D}}\log\left(\sqrt{\frac{\mathbb{P}^\pi_\nu(o)}{\mathbb{P}^\pi_{\nu^*}(o)}}\right)\right)}{\exp\left(|\mathcal{D}|\cdot\log\mathbb{E}_{o\sim\mathbb{P}^\pi_{\nu^*}}\left[\sqrt{\frac{\mathbb{P}^\pi_\nu(o)}{\mathbb{P}^\pi_{\nu^*}(o)}}\right]\right)}\right]\exp(-\beta)$$

$$= \mathbb{E}_{\nu^*}\left[\frac{\Pi_{o\in\mathcal{D}}\sqrt{\frac{\mathbb{P}^\pi_\nu(o)}{\mathbb{P}^\pi_{\nu^*}(o)}}}{\mathbb{E}_{\tau\sim\mathbb{P}^\pi_{\theta^*}}\left[\sqrt{\frac{\mathbb{P}^\pi_\theta(\tau)}{\mathbb{P}^\pi_{\theta^*}(\tau)}}\right]^{|\mathcal{D}|}}\right]\exp(-\beta) = \exp(-\beta),$$

where in the last line, we used the independent property of samples. Thus, again by setting $\beta=\log(M/\delta)$, with probability at least $1-\eta$, we have

$$|\mathcal{D}|\cdot D^2_{\mathrm{H}}(\mathbb{P}^\pi_\nu, \mathbb{P}^\pi_{\nu^*}) \leq -\frac{1}{2}\sum_{o\in\mathcal{D}}\log\left(\frac{\mathbb{P}^\pi_\nu(o)}{\mathbb{P}^\pi_{\nu^*}(o)}\right) + \beta$$

$$= -\frac{1}{2}\sum_{o\in\mathcal{D}}\log\left(\frac{\mathbb{P}^\pi_\nu(o)}{\mathbb{P}^\pi_{\nu^*}(o)}\right) + \frac{1}{2}\sum_{o\in\mathcal{D}}\log\left(\frac{\mathbb{P}^\pi_\nu(o)}{\mathbb{P}^\pi_{\nu^*}(o)}\right) + \beta,$$

for all $k\in[K]$ and $\nu\in\mathbb{M}$. Now we can apply Lemma A.2, and finally have

$$D^2_{\mathrm{H}}(\mathbb{P}^\pi_\nu, \mathbb{P}^\pi_{\nu^*}) \leq \frac{1}{2|\mathcal{D}|}\left(-\sum_{o\in\mathcal{D}}\log\left(\frac{\mathbb{P}^\pi_\nu(o)}{\mathbb{P}^\pi_{\nu^*}(o)}\right) + 3\beta\right).$$

$\qquad\square$

# C. Additional material

## C.1. Greedy algorithm

Algorithm 5 provides the pseudocode of a tractable procedure to compute the greedy test for Algorithm 4 through a $\lambda/4$-discretization of the space of thresholds $b$.

---

**Algorithm 5** Greedy Test

---

1: **input** set of tasks $S$
2: **for** $k \in [K]$ **do**
3:     Define $S^+(b) := \{\nu \in S \mid \hat{\mu}(k) \leq b - \lambda/2\}$
4:     Define $S^-(b) := \{\hat{\nu} \in S \mid \hat{\mu}(k) > b + \lambda/2\}$
5:     Compute $M_k(b) = \max_{b \in [0,1]_{\lambda/4}} \min(|S^+(b)|, |S^-(b)|)$
6: **end for**
7: Extract $(k, b) = \arg\max_{k \in [K]} M_k(b)$
8: **output** greedy test $(\mu(k) \leq b)$

---

## C.2. DT-ECE

Algorithm 6 provides the pseudocode of the DT-ECE algorithm, which implements ECE (Algorithm 1) for a misspecified set of tasks $\hat{\mathbb{M}}$ with a decision tree classifier.

---

**Algorithm 6** Decision Tree – Explicit Classify then Exploit

---

1: **input** set of tasks $\hat{\mathbb{M}}$, decision tree $\texttt{tree}(\hat{\mathbb{M}})$, $N_{\text{cls}} = \frac{2\log(2HD)}{\lambda^2}$
2: Initialize $S_0 = \hat{\mathbb{M}}, t = 0$                                            `Explicit Classify`
3: **while** $|S_t| > 1$ **do**
4:     Extract test $(\mu_k \leq b) = \texttt{tree}(S_t)$
5:     $\mathcal{D}_t \leftarrow N_{\text{cls}}$ i.i.d. samples drawn with $\pi_t = k$
6:     **if** $\frac{1}{N_{\text{cls}}} \sum_{r \in \mathcal{D}_t} r \leq b$ **then**
7:         Get $S_{t+1} \leftarrow \texttt{tree}(S_t, \text{true})$
8:     **else**
9:         Get $S_{t+1} \leftarrow \texttt{tree}(S_t, \text{false})$
10:     **end if**
11: **end while**
12: Extract the classified task $m^* \in S_t$ and execute $\hat{\pi}^*(x) = \arg\max_{\pi \in \Pi} \hat{\nu}_{m^*}(x, k)$ for the remaining steps     `Exploit`

---

