# OpenReview forum: "A Classification View on Meta Learning Bandits"
_ICML.cc/2025/Conference — ICML 2025 poster_

### Official Review · Reviewer_hH2x · 2025-02-15

**Overall Recommendation:** 3

**Summary:**

This paper studies a meta-learning approach to multi-armed bandits (MAB), where multiple bandit instances (tasks) are drawn from an unknown prior distribution. The key contribution is formulating meta-learning bandits as a classification problem, leveraging a novel complexity measure called the classification coefficient.

**Claims And Evidence:**

Yes, most claims made in the submission are supported. However,

1.  There is no lower bound analysis to show that the classification coefficient is the best or tightest complexity measure for this setting. The theoretical justification for why this measure is superior to alternative complexity metrics (e.g., information gain in latent bandits, KL divergence of reward distributions) is missing.

**Essential References Not Discussed:**

N/A

**Experimental Designs Or Analyses:**

1. The experiments only use synthetic datasets, which are useful but do not demonstrate real-world applicability. The task distributions and separability conditions are artificial, making it unclear how well the method generalizes to practical applications.

2. The paper does not test real-world recommendation, healthcare, or RL-based MAB tasks, where meta-learning could provide significant benefits.

3. The theoretical results depend on task separability $\lambda$, but the paper does not systematically evaluate how different levels of
$\lambda$ affect regret.

**Methods And Evaluation Criteria:**

The proposed method is well-motivated. However, there is no analysis of how regret scales with increasing task diversity or task ambiguity.

**Other Comments Or Suggestions:**

1. The intuitive explanation of the classification coefficient \( C_\lambda(M) \) is unclear and difficult to understand.

2. The method is introduced in a straightforward manner, lacking motivational analysis, which makes it feel like a direct application of classification methods.

3. Additionally, there is insufficient experimental visualization, such as task classification processes and decision boundaries.

**Other Strengths And Weaknesses:**

Other Strengths And Weaknesses:

1. The paper introduces the Explicit Classify then Exploit (ECE) algorithm, which consists of two phases: a. The algorithm first identifies the task by collecting data from different arms and using a decision-tree-based classification method. b. Exploitation Phase: Once the task is classified, the algorithm uses the optimal arm selection strategy corresponding to that task, minimizing exploration regret.

2. The paper introduces a classification-based perspective for meta-bandits, which is novel compared to traditional Bayesian or clustering-based approaches.

Other Weaknesses:

1. ECE relies on decision trees for classification, which may not scale well in high-dimensional settings or when task distributions are complex.

2. No ablation studies to understand the effect of task separability $\lambda$ or decision tree depth on classification accuracy and regret.

**Questions For Authors:**

1. No discussion on what happens when tasks overlap—does misclassification degrade performance significantly?

2. The algorithm assumes a fixed set of known tasks—how does it generalize to new, unseen tasks?

3. Is $C_{\lambda}(M)$ provably the best complexity measure for task classification in meta-bandits, or could a better bound be derived?

**Relation To Broader Scientific Literature:**

The approach aligns with previous work in latent bandits and meta-RL, where task identification plays a crucial role. Meta-learning in bandits aims to reuse knowledge from past tasks to improve future decision-making. Alternative approaches, such as latent variable models, clustering, or embedding-based meta-learning, are not explored.

**Theoretical Claims:**

The high-level proof strategy follows standard approaches: First bound the number of rounds required for correct classification; Then analyze regret contributions from both classification and exploitation phases. Mostly correct, but the paper does not provide a lower bound to show whether this regret bound is tight or optimal.  And the paper does not analyze cases where tasks are not well-separated, which could impact the validity of theoretical guarantees.

---

> ### Author Rebuttal · Authors · 2025-04-01
>
> We thank the reviewer for their insightful comments. We provide below detailed replies.
>
> **Questions for authors**
>
>    1. Our analysis and experiments consider overlapping tasks [for “overlap”, we mean the bandits may have the same, or similar, reward distribution for some of the arms/contexts. If the reviewer was thinking of a different kind of overlap, we kindly ask them to please elaborate more on this]. The parameter $\lambda$ controls the degree to which at least one arm distribution *does not overlap* in any pair of bandits. Since the algorithm uses $\lambda$ to make the tests statistically robust, it suffers from the same misclassification rate with any $\lambda$, whereas the regret degrades as $O(\lambda^{-2})$ (see Theorem 3.1, 4.4)
>    2. Note that we also consider a setting where tasks are *not known* to the algorithm, which estimates them through samples (Section 4.1). The reviewer is right that the set of tasks is *fixed* in the paper: We do not provide results for generalization to tasks that are not seen in training. Arguably, generalization depends on “how” the test tasks differ from the training tasks. We conjecture that similar results could be obtained for the *lenient regret* (see https://arxiv.org/abs/2008.03959) when the test task is an $\epsilon$-perturbation of a training task. For more general discrepancies between training and test, the algorithm shall be re-designed to allow for additional test-time exploration, which is a nice direction for future works.
>    3. We proved that $C_{\lambda} (\mathbb{M})$ captures the complexity of an instance of our setting with both a lower bound (Theorem 3.3) and an upper bound (Theorem 3.1).  This implies that $C_{\lambda} (\mathbb{M})$ is *as good as any other* complexity measure for this *specific* setting: Finite collection of separated bandits. For the latter setting, there might be other complexity measures that are equally good (we are not aware of). In other settings, there might be better complexity measures.
>
> **Lower bound**
>
> We actually provide a lower bound where a factor $C_\lambda (\mathbb{M})$ appears explicitly (see Theorem 3.3 and discussion thereafter). We thank the reviewer for making us realize that the lower bound is not highlighted enough. We will make this result more central in the updated manuscript.
>
> **Ablation study on the effect of separability**
>
> Reviewer is mentioning a few times that an analysis of the effect of $\lambda$ on empirical results is missing. We hear their concern: We will report in the Appendix more results for different $\lambda$ values. What we noticed from our experiments is the same effect that one can see by squinting at Figure 4 (a, b). In 4a, $\lambda$ is 0.4 and the elbow in the DT-ECE is very sharp. In 4b, $\lambda$ is 0.04 and the elbow is a lot smoother. With smaller $\lambda$ it takes more time to traverse the tree, both because each test requires more samples and the tree might be deeper.
>
> **Experimental design**
>
> We agree that experiments on real-world data better supports real-world applicability. We will add to the manuscript an experiment on movie recommendations with the MovieLens dataset (https://grouplens.org/datasets/movielens/), on the same line of the MovieLens experiment in Hong et al 2020a (see response to R. 1Rko "experimental design").
>
> **Other weaknesses:**
>
> *“ECE relies on decision trees for classification, which may not scale well in high-dimensional settings or when task distributions are complex”* We recall that the regret of ECE is not affected by the dimensionality of the problem (see Theorem 3.1). However, we believe the Reviewer means something else here: The additional complexity of computing and storing the decision tree. The computational complexity is polynomial in both $d$ (dimensionality) and $M$ (size of the set of tasks), where the factors of $d$ disappear when the tasks are known. The space complexity is $M 2^{C^* (\mathbb{M})} \leq M 2^M$, which may become intractable for a large set of tasks. In those cases, the tree can be traversed *online* without pre-computing it (like in Algorithm 1) with negligible additional space complexity. We will add space complexity and considerations on balancing interpretability and memory requirements in the updated manuscript.
>
> **Intuitive explanation of coefficient $C_\lambda (\mathbb{M})$:**
>
> We thank the reviewer for pointing out that the definition of the classification coefficient is difficult to understand. See the comment for all the reviewers in the response to Reviewer HVdH for an intuitive explanation.
>
> **Insufficient experimental visualizations:**
>
> Following the Reviewer’s suggestion, we will add visualizations of the decision trees learned by our algorithm in the Appendix. As an example, we provide the one for Figure 4a at https://anonymous.4open.science/r/anon_icml_rebuttal-90C9/empirical_tree.pdf. Especially, note the short depth of the tree even for a hard instance that maximizes $C_\lambda (\mathbb{M}) = M$.

---

### Official Review · Reviewer_HVdH · 2025-03-14

**Overall Recommendation:** 4

**Summary:**

The authors address a meta-linear latent contextual bandit setting, where there are M total possible bandit settings, and where there is separation between these M settings. Authors thus propose a classification view that leverages this separation: first, classify the test task as one of M settings; then, perform the greedy policy assuming the test task is correctly classified ("ECE": Explicit Classify then Exploit). Authors prove properties about the proposed method when assuming full knowledge of the bandit settings, including a regret bound that uses a novel measure of complexity. Authors also propose a practical implementation of ECE with decision trees. The resulting exploration policy is interpretable, which can be an important feature in practice. Authors also prove properties of the practical method. Authors demonstrate that the performance of the proposed method is on par with versions of TS and UCB, which are not interpretable.

**Claims And Evidence:**

Theoretical claims:
- Theorems and lemmas have proofs in the appendix. Due to limited time, from spot checking, Theorem 3.1 seems to have no (serious) issues.
- Occasionally claims are made casually in the text that cite another work, but readers could benefit from a proof sketch in the appendix (e.g. Eq 5).

Empirical:
- The proposed method performs on par with versions of TS and UCB: this looks fine to me.

**Essential References Not Discussed:**

There are many papers that are tangentially related that I am familiar with (e.g. other meta-bandit papers) but I wouldn't consider any of them to be essential for discussion.

**Experimental Designs Or Analyses:**

Yes, I read over the experiments section and it looks sensible.

**Methods And Evaluation Criteria:**

Methods:
- The methods (with and without full knowledge of $\mathcal M$) seem very reasonable to me, given the assumed setting.
- I am curious what kinds of realistic settings the assumptions could correspond to, and especially ones where the proposed method should perform particularly well.
- I am curious about what values $N_{cls}$ takes on in practice.

Evaluation:
- The experiments are done on simple settings but the paper contribution is primarily theoretical.
- One could say that because the proposed method is interpretable, unlike existing baselines, it adds value even if it does not outperform. This would be true assuming there do not exist other tree-based exploration policies for this setting.

**Other Comments Or Suggestions:**

- Typos in the abstract, e.g. "asses" and "When human design strategies", and also elsewhere, e.g. "For instance, to minimize the number of times a treatment different from the optimal one is administered to a patient" is not a full sentence.
- Algorithm 1 line 4 should be argmax rather than max
- It seems like / is used to denote set-minus (e.g. line 252, 587), which in my impression is less common than \.
- In Section 3.1 it would improve clarity to explain that the non-optimal arms are such that for $i\neq j$, there is some arm $a$ for which $\mu_i(a)=(1+\lambda)/2$ and $\mu_j(a)=(1-\lambda)/2$. (I assume this is true based on the rest of the section, but it is not explicitly stated.)
- I think line 815 does not follow from line 812, but line 818 is still true. The KL should not disappear on line 815.
- Where does $\hat m_t$ come from in line 593?
- Why is $C(M)\leq C^*(M)$ in (5)?
- In the abstract, part of the motivation is "When human design strategies, they aim for the exploration to be *fast*, since the patient’s health is at stake". This phrasing suggests something more urgent / is measured via something other than better asymptotics. I think this is a minor point.

**Other Strengths And Weaknesses:**

Strengths
- The paper is clearly written.
- The proofs are well-organized and concise.
- The paper does not overpromise or overhype.

Weaknesses
- The linear bandit assumption is strong. However, such assumptions are also common.

**Questions For Authors:**

1. What is $N_{cls}$ in the experiments?

**Relation To Broader Scientific Literature:**

This paper proposes using tree-based classification for interpretable exploration in a meta-learning contextual bandit setting where the total number of settings is fixed ($M$) and the settings have separation. Perhaps I am not that familiar, but I don't often see meta-bandit papers with separation and with tree-based classification.

**Theoretical Claims:**

I checked the proof of Theorem 3.2 for correctness and I did not find (serious) errors.

---

> ### Author Rebuttal · Authors · 2025-04-01
>
> We are glad to hear that the reviewer appreciated our work. We thank them for their thoughtful comments, useful suggestions, and for pointing out typos. We will make use of them to improve the manuscript. We address their questions below.
>
> **Value of $N_{cls}$ in the experiments**
>
> We used $N_{cls} = C / \lambda^2$  in all the experiments, where $C$ is a constant. We set $C = 4$ after trying $C = 2, 4, 8, 16$ (results are not particularly sensitive to this value). Moreover, we implemented an adaptive sampling strategy that checks the confidence of the test at each sample, so that the sampling ends whenever the desired confidence is reached or $N_{cls}$ samples have been taken. The adaptive sampling also didn’t affect results substantially, but it makes the algorithm slightly more efficient in some settings. We thank the reviewer for pointing out that this information is missing in the text. We will add a specific section with experimental details to the Appendix.
>
> **Other comments:**
>   - Why is $C (M) < C^* (M)$ in (5)?
> Thank you for the opportunity to clarify. First, we provide below an detailed explanation of $C (M)$, which can help to understand the proof. Proof: This is a standard argument in decision trees that $C(M)$ is a lower bound to $C^* (M)$. Consider any subset of hypotheses $S$, and when starting from this subset, let the maximum number of hypotheses that we can rule out each round is at most $N$. Then, the depth of any deterministic decision trees with the subset $S$ must be at least $S/N$. Since $C^* (M)$ is the depth of deterministic trees for a larger set of hypothesis, $S/N < C^*(M)$. This holds for all subsets of $M$, hence $C(M) < C^*(M)$.
> Following reviewer’s suggestion, we will add this sketch of the proof of eq. 5 in the Appendix.
>   - *In Section 3.1 it would improve clarity to explain that the non-optimal arms are such that for $i \neq j$ for which $\mu_i (a) = (1 + \lambda) / 2$ and $\mu_j (a) = (1 - \lambda) / 2$ (I assume this is true based on the rest of the section, but it is not explicitly stated.)*
> The reviewer is right! This is implied by the separation assumption (Asm. 1) but we agree with the reviewer it is very implicit. We will explicitly mention this in the text.
>
>
> ------
>
> # **For all the Reviewers**
>
> **Intuitive explanation of $C_\lambda (\mathbb{M})$**
>
> Based on all the reviews, we understand that our exposition of $C_{\lambda} (M)$ could be improved with more intuition. We want to provide an intuitive explanation of its meaning here. We hope that the additional clarity will make the reviewers better appreciate this important contribution.
>
> Informally, the value of $C_\lambda (\mathbb{M})$ measures how much information we gain from a single split of the worst-case node of the decision tree (small value means more information). To see this, let us look at the math:
>
> $$C_\lambda (\mathbb{M}) := \max_{S \in 2^{[M]}} \min_{\pi \in [K]} \max_{i \in S} \frac{|S|}{|S_{\lambda}^\pi (i)|}$$
> Let us unpack each term. $S$ is a set of candidate bandits and $i$ stands for the true bandit within $S$. $\pi$ is an arm we *test* on the true bandit. $S_{\lambda}^\pi$ is the subset of $S$ that we can eliminate from the set of candidates by pulling $\pi$ enough. Thus, the ratio $|S| / |S_{\lambda}^\pi|$ measures the amount of information we can gain from this split. In the extreme cases:
> We eliminate a single bandit from $S$, so that $C_{\lambda} (\mathbb{M}) = M$ is large (little information gained)
> We eliminate half of the bandits in $S$ and $C_{\lambda} (\mathbb{M}) = M / (M/2) = 2$ is small (a lot of information gained). Note that we cannot reduce $C_{\lambda} (\mathbb{M})$ by eliminating more than half of the candidates, as $i$ is chosen worst-case to select the largest set after the split.
>
> We will add this intuitive explanation together with extreme cases and visualizations in the updated manuscript.

---

### Official Review · Reviewer_1Rko · 2025-03-14

**Overall Recommendation:** 2

**Summary:**

This paper presents a novel classification - based approach to meta - learning bandits. Contextual multi - armed bandits are widely used for sequential decision - making, but common bandit algorithms have issues like high regret and lack of interpretability.

The authors consider a meta - learning setting of latent bandits with a separation condition. They introduce a classification - coefficient to measure the complexity of the learning problem. The Explicit Classify then Exploit (ECE) algorithm is proposed, which classifies the test task and exploits the optimal policy of the classified task. To make the algorithm more practical, the Decision Tree ECE (DT - ECE) is developed. It is robust to misspecifications, only accesses samples from the context distribution, and provides an interpretable exploration plan. Experiments show that DT - ECE performs well compared to current bandit approaches.

In conclusion, this classification view offers a new way to design interpretable and efficient exploration plans, and may inspire future research in more general problem settings.

**Claims And Evidence:**

While the authors aim to enhance the interpretability of bandit algorithms, their motivation lacks sufficient depth. The claim that bandit algorithms lack interpretability is controversial, as these algorithms, grounded in probability theory with well - defined mathematical assumptions, already possess a form of inherent interpretability. The paper fails to adequately justify the need for further interpretability. To strengthen the motivation, more real-world examples are essential.

**Essential References Not Discussed:**

Qi Y, Ban Y, Wei T, et al. Meta-learning with neural bandit scheduler[J]. Advances in Neural Information Processing Systems, 2023, 36: 63994-64028.

Sharaf A, Daumé III H. Meta-learning effective exploration strategies for contextual bandits[C]//Proceedings of the AAAI Conference on Artificial Intelligence. 2021, 35(11): 9541-9548.

Xie M, Yin W, Xu H. AutoBandit: A Meta Bandit Online Learning System[C]//IJCAI. 2021: 5028-5031.

**Experimental Designs Or Analyses:**

The experimental datasets are simplistic, consisting entirely of synthetic data with relatively fixed parameters. This simplicity raises concerns about the generalizability of your findings. Real world problems are often complex and variable, and the lack of diverse and realistic datasets means that it's difficult to assess how your proposed methods would perform in practical scenarios.

The absence of scalability experiments is a significant drawback. The super-parameters of setting are fixed.

Regarding the comparison with baseline methods, limiting the comparison to only one or two baselines is insufficient. There are many established methods in the meta bandit literature, and not comparing your approach with a wider range of them makes it difficult to position your work within the existing research landscape. A more comprehensive comparison would provide a better understanding of the relative strengths and weaknesses of your proposed algorithms.

**Methods And Evaluation Criteria:**

Considering only regret might be insufficient. In real-world applications, other factors like space complexity, which will be worse due to the tree, could also be important evaluation aspects that are not fully explored.

**Other Comments Or Suggestions:**

None

**Other Strengths And Weaknesses:**

The paper is replete with a plethora of symbols, which, while essential for the mathematical rigor of the research, lack sufficient explanation of their importance. For example, the \(C_{\lambda}(\mathbb{M})\). Why it is a core contribution? Explanation for the meaningful is better to present the symbols.

**Questions For Authors:**

None

**Relation To Broader Scientific Literature:**

Bandit study is related to many ML fields.

**Theoretical Claims:**

The paper's assumption of a stochastic environment when optimizing multiple candidate bandit instances is a flaw. In reality, these bandits often operate in non-stationary conditions. Each bandit might be at a different stage of convergence, leading to diverse current performances. Could authors explain this point?

---

> ### Author Rebuttal · Authors · 2025-04-01
>
> We want to thank the reviewer for their feedback. We are replying to their comments below.
>
> **Claims and evidence**
>
> Interpretability is a crucial motivation of our paper and we want to make sure we are on the same page with the reviewer on this. Especially:
>   - **Interpretability of bandit algorithms**: Translating a notion from supervised learning to the bandit setting, we say that exploration is interpretable if it can be represented through a decision tree with *small depth* (see Bressan et al. 2024 https://proceedings.mlr.press/v247/bressan24a Def. 2). Our work is the first to establish bounds on the depth of the decision tree for exploration (Lemma 4.3)
>   - **UCB and TS are interpretable**: UCB and TS may also be represented through a decision tree, but its depth can be as large as the number of rounds. According to this notion UCB and TS are *not interpretable*. See an example of how our algorithm builds a compact decision tree (https://anonymous.4open.science/r/anon_icml_rebuttal-90C9/empirical_tree.pdf from the experiment of Figure 4a) whereas an analogous tree for UCB or TS exploration is too large to be visualized
>
> **Methods and evaluation criteria**
>
> Great point! Storing the full decision tree requires additional $M 2^{C^* (\mathbb{M})} \leq M 2^M$ space in the worst-case, which may be impractical in applications where the number of tasks $M$ is huge. In those cases, the tree can be traversed *online* without pre-computing it (like in Algorithm 1) with negligible additional space complexity. We will add space complexity and considerations on balancing interpretability and memory requirements in the updated manuscript.
>
> **Theoretical claims**
>
> “Each bandit might be at a different stage of convergence” we are not sure we understand the reviewer’s concern here. Note that we interact with only one bandit at test time. Moreover, stationarity is orthogonal to whether the reward distributions are stochastic or adversarial. We target the stationary stochastic setting, just like in the previous literature of latent bandits (with the exception of Hong et al 2020b), aiming for fast and interpretable exploration. This makes the approach *more* practical (than previous works) in some applications, e.g., the candidate bandits represent user “types” in a recommendation system: It is often reasonable to assume that a user doesn’t change “type” during the interaction. We agree with the Reviewer that some other applications are inherently non-stationary and extending our results is a non-trivial direction for future works.
>
> **Experimental design**
>
> While we developed a method intended for practical settings, we remark that the core contribution of the paper is conceptual. However, to  support practicality of our method, we will add an additional experiment in which we compute the decision tree on the MovieLens dataset (https://grouplens.org/datasets/movielens/), similarly to Hong et al 2020a (see their Sec 5.2). Users are first clustered into types, based on historical data. Then, the decision tree that we learn essentially prescribes what movies to recommend in order to classify the type of an unknown user, to then maximize their reward by providing recommendations specialized for the classified type.
>   - **Super-parameters are fixed.** We considered different values for the separation parameter, number of arms and bandits. Is there any other parameter the reviewer would like to see explored?
>   - **Baselines.** The aim of the experiments is not to outperform any prior bandit algorithm, as our is the first method that is *interpretable* (in the sense above) and thus competes in a different field. If the reviewer thinks there is a baseline that is relevant to the latter field, we will be happy to include a comparison.
>
> **Essential references**
>
> Thanks for pointing out additional relevant work. We will be happy to include them in the paper with appropriate discussion. Qi et al and Xie et al are interesting but tackle orthogonal problems. S&D 2021 meta learn exploration for contextual bandits, which is similar in nature to our setting. Their algorithm, MELEE, trains exploration by imitating an expert (optimal exploration) in simulation. There are crucial differences:
>   - We do not assume access to an expert during training, but only to simulators
>   - MELEE is not interpretable according to our notion (see above), as we cannot bound the depth of a decision tree representing its policy
>   - MELEE’s regret goes to zero asymptotically, but the regret rate is implicit in their result
>
> **Other weaknesses**
>
> Thanks for letting us know that the symbols are sometimes hard to grasp. We will make a careful revision of all the symbols and to convey enough intuition. See the comment for all the reviewers in the response to Reviewer HVdH for an intuitive explanation of the coefficient $C_{\lambda} (M)$.

---

### Decision · Program_Chairs · 2025-05-01

**Decision:**

Accept (poster)

**Comment:**

This paper proposes a classification-based approach to meta-learning in contextual bandits, introducing an interpretable decision-tree-based algorithm and a novel complexity measure, the classification coefficient $C_\lambda(\mathbb{M})$, to characterize regret.

Reviewers raised concerns about the limited empirical validation and questioned the necessity of the new interpretability framing. However,  authors provided clear responses, offered intuitive explanations of theory, and committed to adding real-world experiments and improved exposition. the paper’s strengths lie in its conceptual novelty, theory, and formal treatment of interpretability.

I encourage the authors to incorporate the promised experimental additions and further clarify technical concepts to strengthen the final version.